# A systematic review and multivariate meta-analysis of the physical and mental health benefits of touch interventions

Receiving touch is of critical importance, as many studies have shown that touch promotes mental and physical well-being. We conducted a pre-registered (PROSPERO: CRD42022304281) systematic review and multilevel meta-analysis encompassing 137 studies in the meta-analysis and 75 additional studies in the systematic review ($n$ = 12,966 individuals, search via Google Scholar, PubMed and Web of Science until 1 October 2022) to identify critical factors moderating touch intervention efficacy. Included studies always featured a touch versus no touch control intervention with diverse health outcomes as dependent variables. Risk of bias was assessed via small study, randomization, sequencing, performance and attrition bias. Touch interventions were especially effective in regulating cortisol levels (Hedges' $g$ = 0.78, 95% confidence interval (CI) 0.24 to 1.31) and increasing weight (0.65, 95% CI 0.37 to 0.94) in newborns as well as in reducing pain (0.69, 95% CI 0.48 to 0.89), feelings of depression (0.59, 95% CI 0.40 to 0.78) and state (0.64, 95% CI 0.44 to 0.84) or trait anxiety (0.59, 95% CI 0.40 to 0.77) for adults. Comparing touch interventions involving objects or robots resulted in similar physical (0.56, 95% CI 0.24 to 0.88 versus 0.51, 95% CI 0.38 to 0.64) but lower mental health benefits (0.34, 95% CI 0.19 to 0.49 versus 0.58, 95% CI 0.43 to 0.73). Adult clinical cohorts profited more strongly in mental health domains compared with healthy individuals (0.63, 95% CI 0.46 to 0.80 versus 0.37, 95% CI 0.20 to 0.55). We found no difference in health benefits in adults when comparing touch applied by a familiar person or a health care professional (0.51, 95% CI 0.29 to 0.73 versus 0.50, 95% CI 0.38 to 0.61), but parental touch was more beneficial in newborns (0.69, 95% CI 0.50 to 0.88 versus 0.39, 95% CI 0.18 to 0.61). Small but significant small study bias and the impossibility to blind experimental conditions need to be considered. Leveraging factors that influence touch intervention efficacy will help maximize the benefits of future interventions and focus research in this field.

The sense of touch has immense importance for many aspects of our life. It is the first of all the senses to develop in newborns[1] and the most direct experience of contact with our physical and social environment[2]. Complementing our own touch experience, we also regularly receive touch from others around us, for example, through consensual hugs, kisses or massages[3].

The recent coronavirus pandemic has raised awareness regarding the need to better understand the effects that touch—and its

✉e-mail: julian.packheiser@rub.de

reduction during social distancing—can have on our mental and physical well-being. The most common touch interventions, for example, massage for adults or kangaroo care for newborns, have been shown to have a wide range of both mental and physical health benefits, from facilitating growth and development to buffering against anxiety and stress, over the lifespan of humans and animals alike[4]. Despite the substantial weight this literature gives to support the benefits of touch, it is also characterized by a large variability in, for example, studied cohorts (adults, children, newborns and animals), type and duration of applied touch (for example, one-time hug versus repeated 60-min massages), measured health outcomes (ranging from physical health outcomes such as sleep and blood pressure to mental health outcomes such as depression or mood) and who actually applies the touch (for example, partner versus stranger).

A meaningful tool to make sense of this vast amount of research is through meta-analysis. While previous meta-analyses on this topic exist, they were limited in scope, focusing only on particular types of touch, cohorts or specific health outcomes (for example, refs. 5,6). Furthermore, despite best efforts, meaningful variables that moderate the efficacy of touch interventions could not yet be identified. However, understanding these variables is critical to tailor touch interventions and guide future research to navigate this diverse field with the ultimate aim of promoting well-being in the population.

In this Article, we describe a pre-registered, large-scale systematic review and multilevel, multivariate meta-analysis to address this need with quantitative evidence for (1) the effect of touch interventions on physical and mental health and (2) which moderators influence the efficacy of the intervention. In particular, we ask whether and how strongly health outcomes depend on the dynamics of the touching dyad (for example, humans or robots/objects, familiarity and touch directionality), demographics (for example, clinical status, age or sex), delivery means (for example, type of touch intervention or touched body part) and procedure (for example, duration or number of sessions). We did so separately for newborns and for children and adults, as the health outcomes in newborns differed substantially from those in the other age groups. Despite the focus of the analysis being on humans, it is widely known that many animal species benefit from touch interactions and that engaging in touch promotes their well-being as well[7]. Since animal models are essential for the investigation of the mechanisms underlying biological processes and for the development of therapeutic approaches, we accordingly included health benefits of touch interventions in non-human animals as part of our systematic review. However, this search yielded only a small number of studies, suggesting a lack of research in this domain, and as such, was insufficient to be included in the meta-analysis. We evaluate the identified animal studies and their findings in the discussion.

## Results

### Touch interventions have a medium-sized effect

The pre-registration can be found at ref. 8. The flowchart for data collection and extraction is depicted in Fig. 1.

For adults, a total of $n = 2,841$ and $n = 2,556$ individuals in the touch and control groups, respectively, across 85 studies and 103 cohorts were included. The effect of touch overall was medium-sized ($t(102) = 9.74$, $P < 0.001$, Hedges' $g = 0.52$, 95% confidence interval (CI) 0.42 to 0.63; Fig. 2a). For newborns, we could include 63 cohorts across 52 studies comprising a total of $n = 2,134$ and $n = 2,086$ newborns in the touch and control groups, respectively, with an overall effect almost identical to the older age group ($t(62) = 7.53$, $P < 0.001$, Hedges' $g = 0.56$, 95% CI 0.41 to 0.71; Fig. 2b), suggesting that, despite distinct health outcomes, touch interventions show comparable effects across newborns and adults. Using these overall effect estimates, we conducted a power sensitivity analysis of all the included primary studies to investigate whether such effects could be reliably detected[9]. Sufficient power to detect such effect sizes was rare in individual studies, as investigated

by firepower plots[10] (Supplementary Figs. 1 and 2). No individual effect size from either meta-analysis was overly influential (Cook's $D < 0.06$). The benefits were similar for mental and physical outcomes (mental versus physical; adults: $t(101) = 0.79$, $P = 0.432$, Hedges' $g$ difference of −0.05, 95% CI −0.16 to 0.07, Fig. 2c; newborns: $t(61) = 1.08$, $P = 0.284$, Hedges' $g$ difference of −0.19, 95% CI −0.53 to 0.16, Fig. 2d).

On the basis of the overall effect of both meta-analyses as well as their median sample sizes, the minimum number of studies necessary for subgroup analyses to achieve 80% power was $k = 9$ effects for adults and $k = 8$ effects for newborns (Supplementary Figs. 5 and 6). Assessing specific health outcomes with sufficient power in more detail in adults (Fig. 3a) revealed smaller benefits to sleep and heart rate parameters, moderate benefits to positive and negative affect, diastolic blood and systolic blood pressure, mobility and reductions of the stress hormone cortisol and larger benefits to trait and state anxiety, depression, fatigue and pain. Post hoc tests revealed stronger benefits for pain, state anxiety, depression and trait anxiety compared with respiratory, sleep and heart rate parameters (see Fig. 3 for all post hoc comparisons). Reductions in pain and state anxiety were increased compared with reductions in negative affect ($t(83) = 2.54$, $P = 0.013$, Hedges' $g$ difference of 0.31, 95% CI 0.07 to 0.55; $t(83) = 2.31$, $P = 0.024$, Hedges' $g$ difference of 0.27, 95% CI 0.03 to 0.51). Benefits to pain symptoms were higher compared with benefits to positive affect ($t(83) = 2.22$, $P = 0.030$, Hedges' $g$ difference of 0.29, 95% CI 0.04 to 0.54). Finally, touch resulted in larger benefits to cortisol release compared with heart rate parameters ($t(83) = 2.30$, $P = 0.024$, Hedges' $g$ difference of 0.26, 95% CI 0.04–0.48).

In newborns, only physical health effects offered sufficient data for further analysis. We found no benefits for digestion and heart rate parameters. All other health outcomes (cortisol, liver enzymes, respiration, temperature regulation and weight gain) showed medium to large effects (Fig. 3b). We found no significant differences among any specific health outcomes.

### Non-human touch and skin-to-skin contact

In some situations, a fellow human is not readily available to provide affective touch, raising the question of the efficacy of touch delivered by objects and robots[11]. Overall, we found humans engaging in touch with other humans or objects to have medium-sized health benefits in adults, without significant differences ($t(99) = 1.05$, $P = 0.295$, Hedges' $g$ difference of 0.12, 95% CI −0.11 to 0.35; Fig. 4a). However, differentiating physical versus mental health benefits revealed similar benefits for human and object touch on physical health outcomes, but larger benefits on mental outcomes when humans were touched by humans ($t(97) = 2.32$, $P = 0.022$, Hedges' $g$ difference of 0.24, 95% CI 0.04 to 0.44; Fig. 4b). It must be noted that touching with an object still showed a significant effect (see Supplementary Fig. 7 for the corresponding orchard plot).

We considered the possibility that this effect was due to missing skin-to-skin contact in human–object interactions. Thus, we investigated human–human interactions with and without skin-to-skin contact (Fig. 4c). In line with the hypothesis that skin-to-skin contact is highly relevant, we again found stronger mental health benefits in the presence of skin-to-skin contact that however did not achieve nominal significance ($t(69) = 1.95$, $P = 0.055$, Hedges' $g$ difference of 0.41, 95% CI −0.00 to 0.82), possibly because skin-to-skin contact was rarely absent in human–human interactions, leading to a decrease in power of this analysis. Results for skin-to-skin contact as an overall moderator can be found in Supplementary Fig. 8.

### Influences of type of touch

The large majority of touch interventions comprised massage therapy in adults and kangaroo care in newborns (see Supplementary Table 1 for a complete list of interventions across studies). However, comparing the different types of touch explored across studies did not reveal significant differences in effect sizes based on touch type,

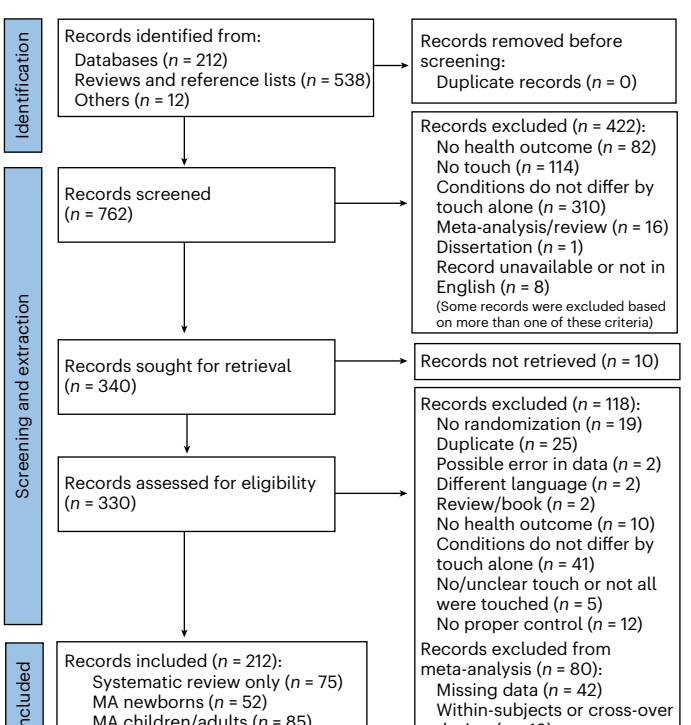

**Fig. 1 | PRISMA 2020 flowchart detailing the identification and screening of identified records for the systematic review and meta-analysis.** Animal outcomes refer to outcomes measured in non-human species that were solely considered as part of a systematic review. Included languages were French, Dutch, German and English, but our search did not identify any articles in French, Dutch or German. MA, meta-analysis.

be it on overall health benefits (adults: $t(101) = 0.11$, $P = 0.916$, Hedges' $g$ difference of 0.02, 95% CI −0.32 to 0.29; Fig. 5a) or comparing different forms of touch separately for physical (massage therapy versus other forms: $t(99) = 0.99$, $P = 0.325$, Hedges' $g$ difference 0.16, 95% CI −0.15 to 0.47) or for mental health benefits (massage therapy versus other forms: $t(99) = 0.75$, $P = 0.458$, Hedges' $g$ difference of 0.13, 95% CI −0.22 to 0.48) in adults (Fig. 5c; see Supplementary Fig. 9 for the corresponding orchard plot). A similar picture emerged for physical health effects in newborns (massage therapy versus kangaroo care: $t(58) = 0.94$, $P = 0.353$, Hedges' $g$ difference of 0.15, 95% CI −0.17 to 0.47; massage therapy versus other forms: $t(58) = 0.56$, $P = 0.577$, Hedges' $g$ difference of 0.13, 95% CI −0.34 to 0.60; kangaroo care versus other forms: $t(58) = 0.07$, $P = 0.947$, Hedges' $g$ difference of 0.02, 95% CI −0.46 to 0.50; Fig. 5d; see also Supplementary Fig. 10 for the corresponding orchard plot). This suggests that touch types may be flexibly adapted to the setting of every touch intervention.

### The role of clinical status

Most research on touch interventions has focused on clinical samples, but are benefits restricted to clinical cohorts? We found health benefits to be significant in clinical and healthy populations (Fig. 6), whether all outcomes are considered (Fig. 6a,b) or physical and mental health outcomes are separated (Fig. 6c,d, see Supplementary Figs. 11 and 12 for the corresponding orchard plots). In adults, however, we found higher mental health benefits for clinical populations compared with healthy ones (Fig. 6c; $t(99) = 2.11$, $P = 0.037$, Hedges' $g$ difference of 0.25, 95% CI 0.01 to 0.49).

A more detailed analysis of specific clinical conditions in adults revealed positive mental and physical health benefits for almost all assessed clinical disorders. Differences between disorders were not found, with the exception of increased effectiveness of touch interventions in neurological disorders (Supplementary Fig. 13).

### Familiarity in the touching dyad and intervention location

Touch interventions can be performed either by familiar touchers (partners, family members or friends) or by unfamiliar touchers (health care professionals). In adults, we did not find an impact of familiarity of the toucher ($t(99) = 0.12$, $P = 0.905$, Hedges' $g$ difference of 0.02, 95% CI −0.27 to 0.24; Fig. 7a; see Supplementary Fig. 14 for the corresponding orchard plot). Similarly, investigating the impact on mental and physical health benefits specifically, no significant differences could be detected, suggesting that familiarity is irrelevant in adults. In contrast, touch applied to newborns by their parents (almost all studies only included touch by the mother) was significantly more beneficial compared with unfamiliar touch ($t(60) = 2.09$, $P = 0.041$, Hedges' $g$ difference of 0.30, 95% CI 0.01 to 0.59) (Fig. 7b; see Supplementary Fig. 15 for the corresponding orchard plot). Investigating mental and physical health benefits specifically revealed no significant differences. Familiarity with the location in which the touch was applied (familiar being, for example, the participants' home) did not influence the efficacy of touch interventions (Supplementary Fig. 16).

### Frequency and duration of touch interventions

How often and for how long should touch be delivered? For adults, the median touch duration across studies was 20 min and the median number of touch interventions was four sessions with an average time interval of 2.3 days between each session. For newborns, the median touch duration across studies was 17.5 min and the median number of touch interventions was seven sessions with an average time interval of 1.3 days between each session.

Delivering more touch sessions increased benefits in adults, whether overall ($t(101) = 4.90$, $P < 0.001$, Hedges' $g = 0.02$, 95% CI 0.01 to 0.03), physical ($t(81) = 3.07$, $P = 0.003$, Hedges' $g = 0.02$, 95% CI 0.01–0.03) or mental benefits ($t(72) = 5.43$, $P < 0.001$, Hedges' $g = 0.02$, 95% CI 0.01–0.03) were measured (Fig. 8a). A closer look at specific outcomes for which sufficient data were available revealed that positive associations between the number of sessions and outcomes were found for trait anxiety ($t(12) = 7.90$, $P < 0.001$, Hedges' $g = 0.03$, 95% CI 0.02–0.04), depression ($t(20) = 10.69$, $P < 0.001$, Hedges' $g = 0.03$, 95% CI 0.03–0.04) and pain ($t(37) = 3.65$, $P < 0.001$, Hedges' $g = 0.03$, 95% CI 0.02–0.05), indicating a need for repeated sessions to improve these adverse health outcomes. Neither increasing the number of sessions for newborns nor increasing the duration of touch per session in adults or newborns increased health benefits, be they physical or mental (Fig. 8b–d). For continuous moderators in adults, we also looked at specific health outcomes as sufficient data were generally available for further analysis. Surprisingly, we found significant negative associations between touch duration and reductions of cortisol ($t(24) = 2.71$, $P = 0.012$, Hedges' $g = −0.01$, 95% CI −0.01 to −0.00) and heart rate parameters ($t(21) = 2.35$, $P = 0.029$, Hedges' $g = −0.01$, 95% CI −0.02 to −0.00).

### Demographic influences of sex and age

We used the ratio between women and men in the single-study samples as a proxy for sex-specific effects. Sex ratios were heavily skewed towards larger numbers of women in each cohort (median 83% women), and we could not find significant associations between sex ratio and overall ($t(62) = 0.08$, $P = 0.935$, Hedges' $g = 0.00$, 95% CI −0.00 to 0.01), mental ($t(43) = 0.55$, $P = 0.588$, Hedges' $g = 0.00$, 95% CI −0.00 to 0.01) or physical health benefits ($t(51) = 0.15$, $P = 0.882$, Hedges' $g = −0.00$, 95% CI −0.01 to 0.01). For specific outcomes that could be further analysed, we found a significant positive association of sex ratio with reductions in cortisol secretion ($t(18) = 2.31$, $P = 0.033$, Hedges' $g = 0.01$, 95% CI 0.00 to 0.01) suggesting stronger benefits in women. In contrast to adults, sex ratios were balanced in samples of newborns (median

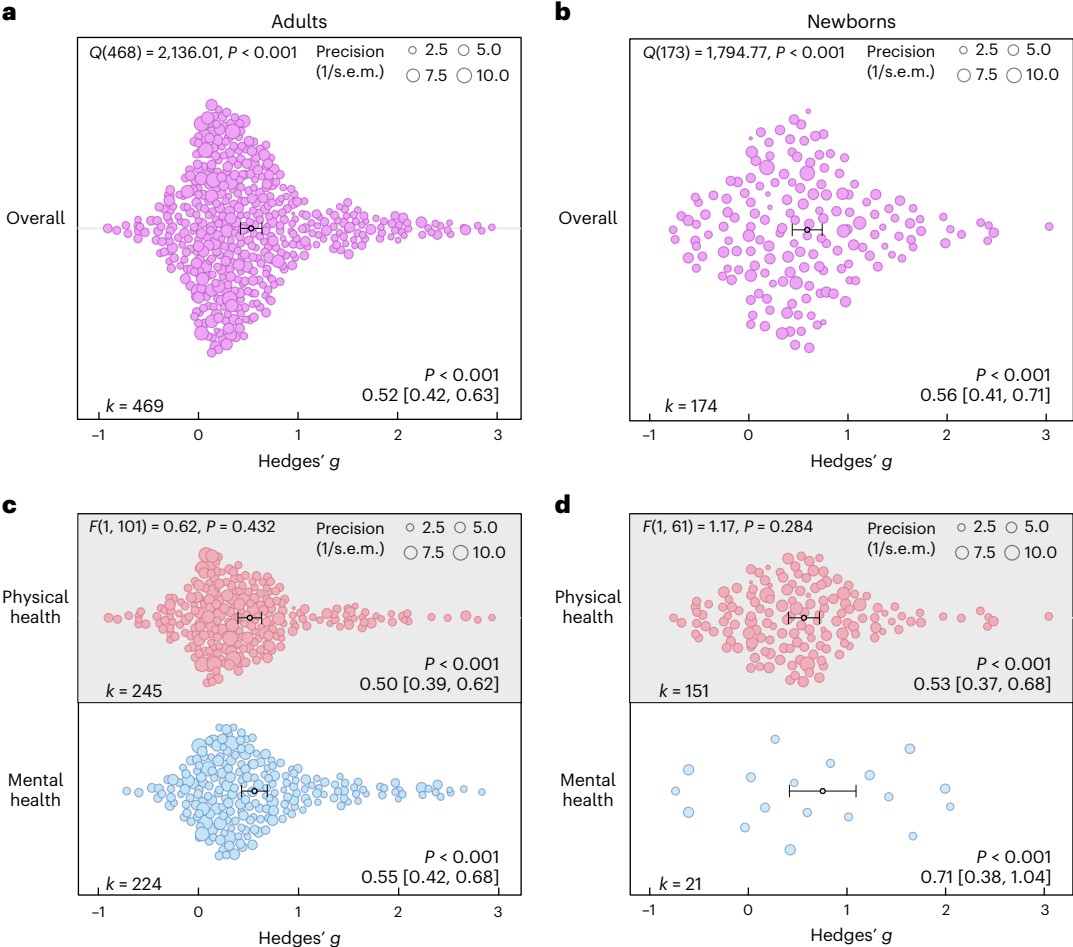

**Fig. 2 | Benefits of touch on physical and mental health. a**, Orchard plot illustrating the overall benefits across all health outcomes for adults/children across 469 in part dependent effect sizes from 85 studies and 103 cohorts. **b**, The same as **a** but for newborns across 174 in part dependent effect sizes from 52 studies and 63 cohorts. **c**, The same as **a** but separating the results for physical versus mental health benefits across 469 in part dependent effect sizes from 85 studies and 103 cohorts. **d**, The same as **b** but separating the results for physical versus mental health benefits across 172 in part dependent effect sizes from 52 studies and 63 cohorts. Each dot reflects a measured effect, and the number of effects (*k*) included in the analysis is depicted in the bottom left. Mean effects and 95% CIs

are presented in the bottom right and are indicated by the central black dot (mean effect) and its error bars (95% CI). The heterogeneity *Q* statistic is presented in the top left. Overall effects of moderator impact were assessed via an *F* test, and post hoc comparisons were done using *t* tests (two-sided test). Note that the *P* values above the mean effects indicate whether an effect differed significantly from a zero effect. *P* values were not corrected for multiple comparisons. The dot size reflects the precision of each individual effect (larger indicates higher precision). Small-study bias for the overall effect was significant (*F* test, two-sided test) in the adult meta-analysis ($F(1, 101) = 21.24$, $P < 0.001$; Supplementary Fig. 3) as well as in the newborn meta-analysis ($F(1, 61) = 5.25$, $P = 0.025$; Supplementary Fig. 4).

---

53% girls). For newborns, there was no significant association with overall ($t(36) = 0.77$, $P = 0.447$, Hedges' $g = -0.01$, 95% CI −0.02 to 0.01) and physical health benefits of touch ($t(35) = 0.93$, $P = 0.359$, Hedges' $g = -0.01$, 95% CI −0.02 to 0.01). Mental health benefits did not provide sufficient data for further analysis.

The median age in the adult meta-analysis was 42.6 years (s.d. 21.16 years, range 4.5–88.4 years). There was no association between age and the overall ($t(73) = 0.35$, $P = 0.727$, Hedges' $g = 0.00$, 95% CI −0.01 to 0.01), mental ($t(53) = 0.94$, $P = 0.353$, Hedges' $g = 0.01$, 95% CI −0.01 to 0.02) and physical health benefits of touch ($t(60) = 0.16$, $P = 0.870$, Hedges' $g = 0.00$, 95% CI −0.01 to 0.01). Looking at specific health outcomes, we found significant positive associations between mean age and improved positive affect ($t(10) = 2.54$, $P = 0.030$, Hedges' $g = 0.01$, 95% CI 0.00 to 0.02) as well as systolic blood pressure ($t(11) = 2.39$, $P = 0.036$, Hedges' $g = 0.02$, 95% CI 0.00 to 0.04).

### Body part
A list of touched body parts can be found in Supplementary Table 1. For the touched body part, we found significantly higher health benefits for

head touch compared with arm touch ($t(40) = 2.14$, $P = 0.039$, Hedges' $g$ difference of 0.78, 95% CI 0.07 to 1.49) and torso touch ($t(40) = 2.23$, $P = 0.031$; Hedges' $g$ difference of 0.84, 95% CI 0.10 to 1.58; Supplementary Fig. 17). Touching the arm resulted in lower mental health compared with physical health benefits ($t(37) = 2.29$, $P = 0.028$, Hedges' $g$ difference of −0.35, 95% CI −0.65 to −0.05). Furthermore, we found a significantly increased physical health benefit when the head was touched as opposed to the torso ($t(37) = 2.10$, $P = 0.043$, Hedges' $g$ difference of 0.96, 95% CI 0.06 to 1.86). Thus, head touch such as a face or scalp massage could be especially beneficial.

### Directionality
In adults, we tested whether a uni- or bidirectional application of touch mattered. The large majority of touch was applied unidirectionally ($k = 442$ of 469 effects). Unidirectional touch had higher health benefits ($t(101) = 2.17$, $P = 0.032$, Hedges' $g$ difference of 0.30, 95% CI 0.03 to 0.58) than bidirectional touch. Specifically, mental health benefits were higher in unidirectional touch ($t(99) = 2.33$, $P = 0.022$, Hedges' $g$ difference of 0.46, 95% CI 0.06 to 0.66).

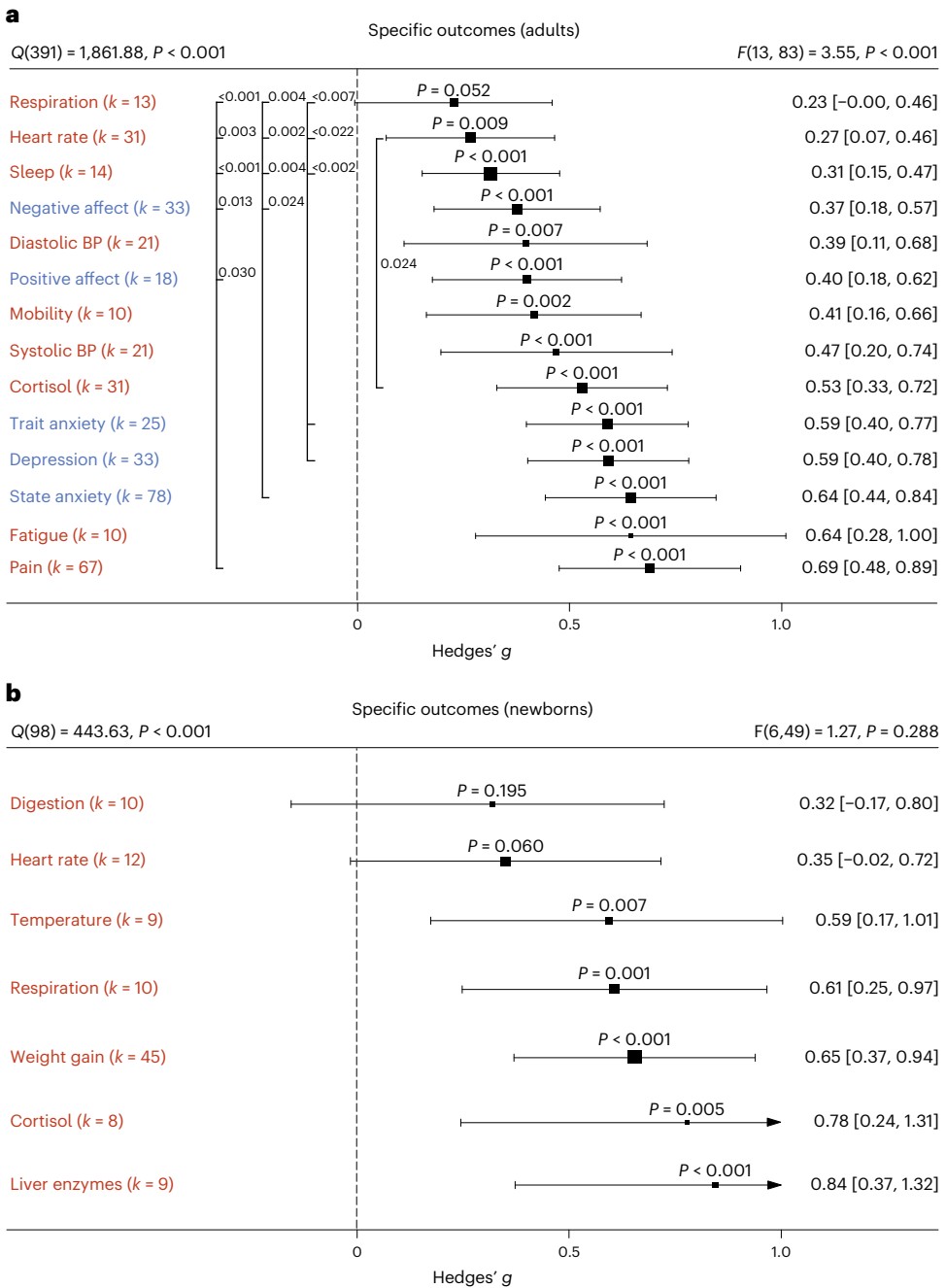

**Fig. 3 | Forest plot for all specific health outcomes with sufficient effects to warrant further analysis. a,b**, Health outcomes in adults analysed across 405 in part dependent effect sizes from 79 studies and 97 cohorts (**a**) and in newborns analysed across 105 in part dependent effect sizes from 46 studies and 56 cohorts (**b**). The type of health outcomes measured differed between adults and newborns and were thus analysed separately. Numbers on the right represent the mean effect with its 95% CI in square brackets and the significance level estimating the likelihood that the effect is equal to zero. Overall effects of moderator impact were assessed via an *F* test, and post hoc comparisons were done using *t* tests (two-sided test). The *F* value in the top right represents a test of the hypothesis that all effects within the subpanel are equal. The *Q* statistic represents the heterogeneity. *P* values of post hoc tests are depicted whenever significant. *P* values above the horizontal whiskers indicate whether an effect differed significantly from a zero effect. Vertical lines indicate significant post hoc tests between moderator levels. *P* values were not corrected for multiple comparisons. Physical outcomes are marked in red. Mental outcomes are marked in blue.

## Study location

For adults, we found significantly stronger health benefits of touch in South American compared with North American cohorts ($t(95) = 2.03$, $P = 0.046$, Hedges' *g* difference of 0.37, 95% CI 0.01 to 0.73) and European cohorts ($t(95) = 2.22$, $P = 0.029$, Hedges' *g* difference of 0.36, 95% CI 0.04 to 0.68). For newborns, we found weaker effects in North American cohorts compared to Asian ($t(55) = 2.28$, $P = 0.026$, Hedges' *g* difference of −0.37, 95% CI −0.69 to −0.05) and European cohorts ($t(55) = 2.36$, $P = 0.022$,

Hedges' *g* difference of −0.40, 95% CI −0.74 to −0.06). Investigating the interaction with mental and physical health benefits did not reveal any effects of study location in both meta-analyses (Supplementary Fig. 18).

## Systematic review of studies without effect sizes

All studies where effect size data could not be obtained or that did not meet the meta-analysis inclusion criteria can be found on the OSF project[12] in the file 'Study_lists_final_revised.xlsx'

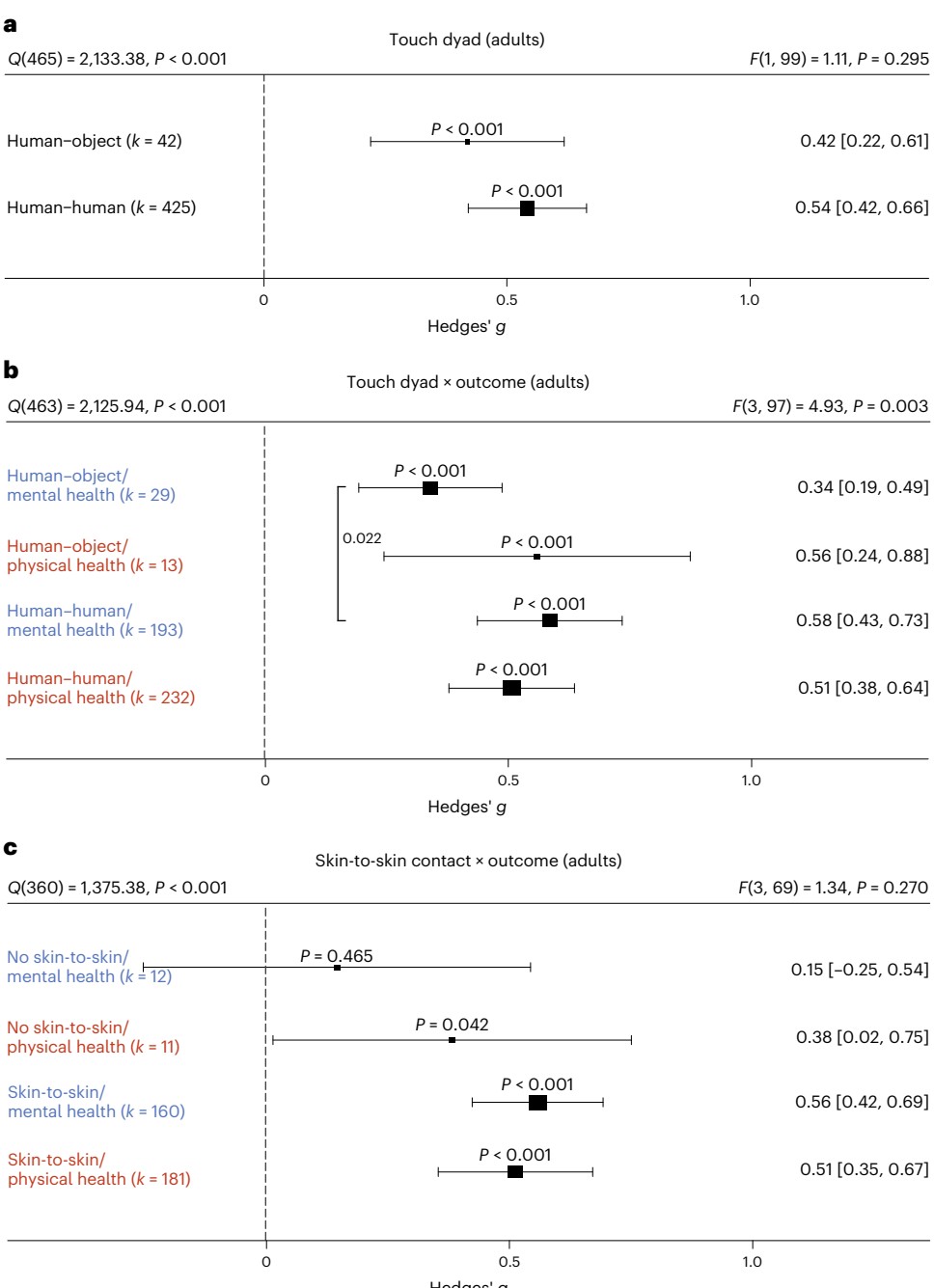

**Fig. 4 | Influence of the touching dyad in adults. a**, Forest plot comparing humans versus objects touching a human on health outcomes overall across 467 in part dependent effect sizes from 85 studies and 101 cohorts. **b**, The same as **a** but separately for mental versus physical health outcomes across 467 in part dependent effect sizes from 85 studies and 101 cohorts. **c**, Results with the removal of all object studies, leaving 406 in part dependent effect sizes from 71 studies and 88 cohorts to identify whether missing skin-to-skin contact is the relevant mediator of higher mental health effects in human–human interactions. Numbers on the right represent the mean effect with its 95% CI in square brackets and the significance level estimating the likelihood that the effect is equal to zero. Overall effects of moderator impact were assessed via an *F* test, and post hoc comparisons were done using *t* tests (two-sided test). The *F* value in the top right represents a test of the hypothesis that all effects within the subpanel are equal. The *Q* statistic represents the heterogeneity. *P* values of post hoc tests are depicted whenever significant. *P* values above the horizontal whiskers indicate whether an effect differed significantly from a zero effect. Vertical lines indicate significant post hoc tests between moderator levels. *P* values were not corrected for multiple comparisons. Physical outcomes are marked in red. Mental outcomes are marked in blue.

(sheet 'Studies_without_effect_sizes'). Specific reasons for exclusion are furthermore documented in Supplementary Table 2. For human health outcomes assessed across 56 studies and $n = 2,438$ individuals, interventions mostly comprised massage therapy ($k = 86$ health outcomes) and kangaroo care ($k = 33$ health outcomes). For datasets where no effect size could be computed, 90.0% of mental health and 84.3% of physical health parameters were positively impacted by touch. Positive impact of touch did not differ between types of touch interventions. These results match well with the observations of the meta-analysis of a highly positive benefit of touch overall, irrespective of whether a massage or any other intervention is applied.

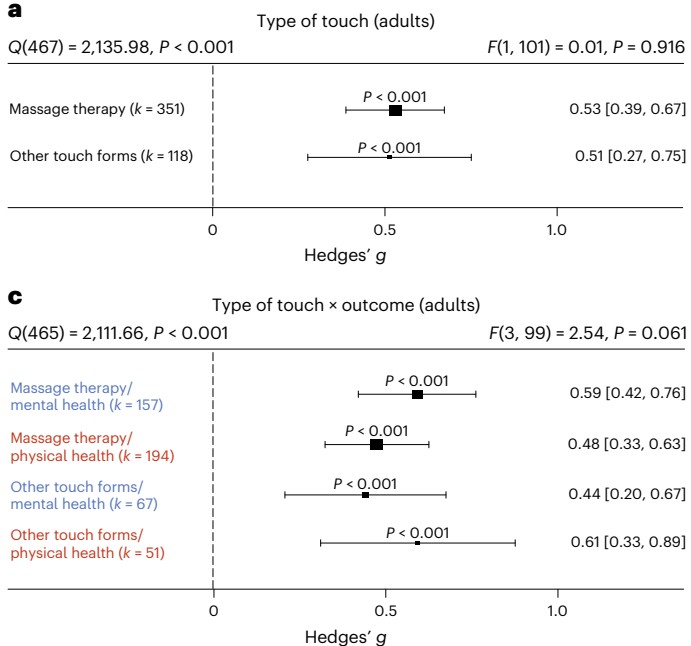

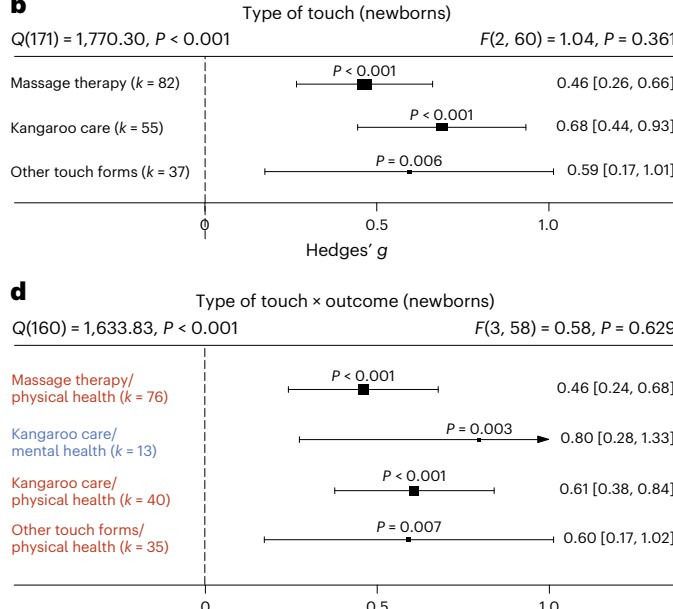

**Fig. 5 | Effect of type of touch. a**, Forest plot of health benefits comparing massage therapy versus other forms of touch in adult cohorts across 469 in part dependent effect sizes from 85 studies and 103 cohorts. **b**, Forest plot of health benefits comparing massage therapy, kangaroo care and other forms of touch for newborns across 174 in part dependent effect sizes from 52 studies and 63 cohorts. **c**, The same as **a** but separating mental and physical health benefits across 469 in part dependent effect sizes from 85 studies and 103 cohorts. **d**, The same as **b** but separating mental and physical health outcomes where possible across 164 in part dependent effect sizes from 51 studies and 62 cohorts. Note that an insufficient number of studies assessed mental health benefits of massage therapy or other forms of touch to be included. Numbers on the right represent the mean effect with its 95% CI in square brackets and the significance level estimating the likelihood that the effect is equal to zero. Overall effects of moderator impact were assessed via an $F$ test, and post hoc comparisons were done using $t$ tests (two-sided test). The $F$ value in the top right represents a test of the hypothesis that all effects within the subpanel are equal. The $Q$ statistic represents heterogeneity. $P$ values of post hoc tests are depicted whenever significant. $P$ values above the horizontal whiskers indicate whether an effect differed significantly from a zero effect. Vertical lines indicate significant post hoc tests between moderator levels. $P$ values were not corrected for multiple comparisons. Physical outcomes are marked in red. Mental outcomes are marked in blue.

We also assessed health outcomes in animals across 19 studies and $n = 911$ subjects. Most research was conducted in rodents. Animals that received touch were rats (ten studies, $k = 16$ health outcomes), mice (four studies, $k = 7$ health outcomes), macaques (two studies, $k = 3$ health outcomes), cats (one study, $k = 3$ health outcomes), lambs (one study, $k = 2$ health outcomes) and coral reef fish (one study, $k = 1$ health outcome). Touch interventions mostly comprised stroking ($k = 13$ health outcomes) and tickling ($k = 10$ health outcomes). For animal studies, 71.4% of effects showed benefits to mental health-like parameters and 81.8% showed positive physical health effects. We thus found strong evidence that touch interventions, which were mostly conducted by humans (16 studies with human touch versus 3 studies with object touch), had positive health effects in animal species as well.

## Discussion

The key aim of the present study was twofold: (1) to provide an estimate of the effect size of touch interventions and (2) to disambiguate moderating factors to potentially tailor future interventions more precisely. Overall, touch interventions were beneficial for both physical and mental health, with a medium effect size. Our work illustrates that touch interventions are best suited for reducing pain, depression and anxiety in adults and children as well as for increasing weight gain in newborns. These findings are in line with previous meta-analyses on this topic, supporting their conclusions and their robustness to the addition of more datasets. One limitation of previous meta-analyses is that they focused on specific health outcomes or populations, despite primary studies often reporting effects on multiple health parameters simultaneously (for example, ref. 13 focusing on neck and shoulder pain and ref. 14 focusing on massage therapy in preterms). To our knowledge, only

ref. 5 provides a multivariate picture for a large number of dependent variables. However, this study analysed their data in separate random effects models that did not account for multivariate reporting nor for the multilevel structure of the data, as such approaches have only become available recently. Thus, in addition to adding a substantial amount of new data, our statistical approach provides a more accurate depiction of effect size estimates. Additionally, our study investigated a variety of moderating effects that did not reach significance (for example, sex ratio, mean age or intervention duration) or were not considered (for example, the benefits of robot or object touch) in previous meta-analyses in relation to touch intervention efficacy[5], probably because of the small number of studies with information on these moderators in the past. Owing to our large-scale approach, we reached high statistical power for many moderator analyses. Finally, previous meta-analyses on this topic exclusively focused on massage therapy in adults or kangaroo care in newborns[15], leaving out a large number of interventions that are being carried out in research as well as in everyday life to improve well-being. Incorporating these studies into our study, we found that, in general, both massages and other types of touch, such as gentle touch, stroking or kangaroo care, showed similar health benefits.

While it seems to be less critical which touch intervention is applied, the frequency of interventions seems to matter. More sessions were positively associated with the improvement of trait outcomes such as depression and anxiety but also pain reductions in adults. In contrast to session number, increasing the duration of individual sessions did not improve health effects. In fact, we found some indications of negative relationships in adults for cortisol and blood pressure. This could be due to habituating effects of touch on the sympathetic

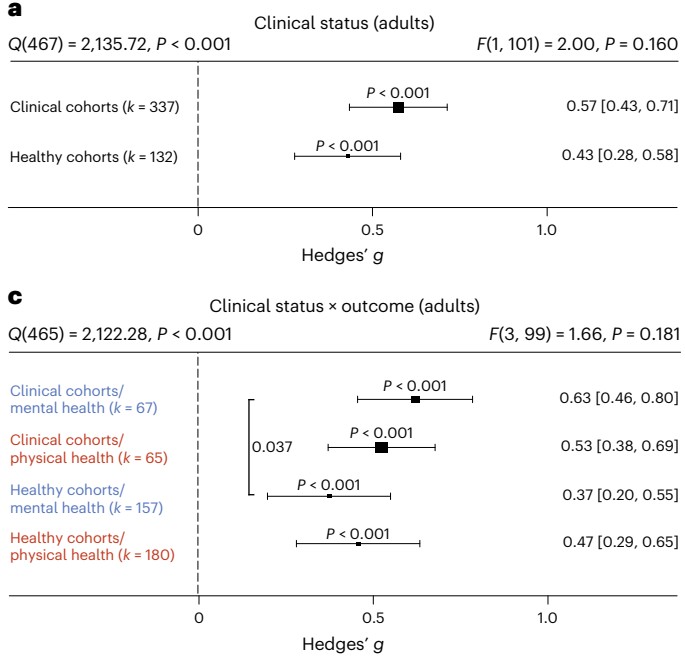

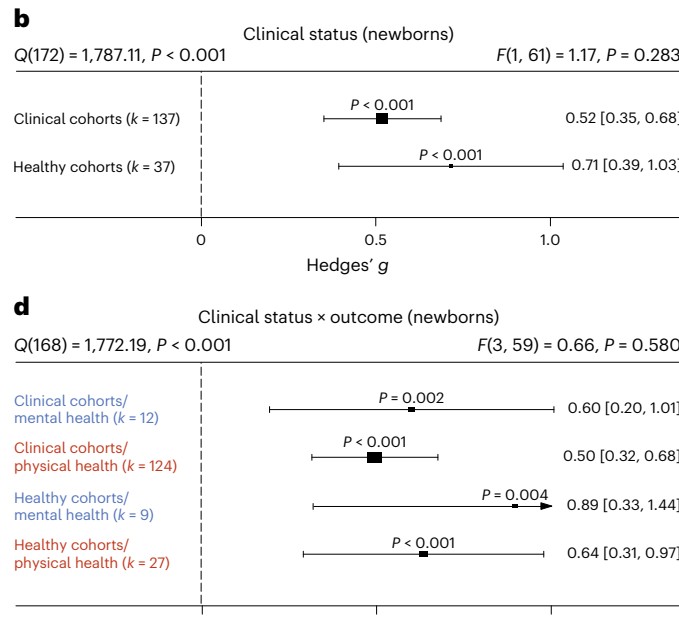

**Fig. 6 | Comparing health benefits for clinical versus healthy cohorts. a**, Health benefits for clinical cohorts of adults versus healthy cohorts of adults across 469 in part dependent effect sizes from 85 studies and 103 cohorts. **b**, The same as **a** but for newborn cohorts across 174 in part dependent effect sizes from 52 studies and 63 cohorts. **c**, The same as **a** but separating mental versus physical health benefits across 469 in part dependent effect sizes from 85 studies and 103 cohorts. **d**, The same as **b** but separating mental versus physical health benefits across 172 in part dependent effect sizes from 52 studies and 63 cohorts. Numbers on the right represent the mean effect with its 95% CI in square brackets and the significance level estimating the likelihood that the effect is equal to zero. Overall effects of moderator impact were assessed via an $F$ test, and post hoc comparisons were done using $t$ tests (two-sided test). The $F$ value in the top right represents a test of the hypothesis that all effects within the subpanel are equal. The $Q$ statistic represents the heterogeneity. $P$ values of post hoc tests are depicted whenever significant. $P$ values above the horizontal whiskers indicate whether an effect differed significantly from a zero effect. Vertical lines indicate significant post hoc tests between moderator levels. $P$ values were not corrected for multiple comparisons. Physical outcomes are marked in red. Mental outcomes are marked in blue.

nervous system and hypothalamic–pituitary–adrenal axis, ultimately resulting in diminished effects with longer exposure, or decreased pleasantness ratings of affective touch with increasing duration[16]. For newborns, we could not support previous notions that the duration of the touch intervention is linked to benefits in weight gain[17]. Thus, an ideal intervention protocol does not seem to have to be excessively long. It should be noted that very few interventions lasted less than 5 min, and it therefore remains unclear whether very short interventions have the same effect.

A critical issue highlighted in the pandemic was the lack of touch due to social restrictions[18]. To accommodate the need for touch in individuals with small social networks (for example, institutionalized or isolated individuals), touch interventions using objects/robots have been explored in the past (for a review, see ref. 11). We show here that touch interactions outside of the human–human domain are beneficial for mental and physical health outcomes. Importantly, object/robot touch was not as effective in improving mental health as human-applied touch. A sub-analysis of missing skin-to-skin contact among humans indicated that mental health effects of touch might be mediated by the presence of skin-to-skin contact. Thus, it seems profitable to include skin-to-skin contact in future touch interventions, in line with previous findings in newborns[19]. In robots, recent advancements in synthetic skin[20] should be investigated further in this regard. It should be noted that, although we did not observe significant differences in physical health benefits between human–human and human–object touch, the variability of effect sizes was higher in human–object touch. The conditions enabling object or robot interactions to improve well-being should therefore be explored in more detail in the future.

Touch was beneficial for both healthy and clinical cohorts. These data are critical as most previous meta-analytic research has focused on individuals diagnosed with clinical disorders (for example, ref. 6). For mental health outcomes, we found larger effects in clinical cohorts. A possible reason could relate to increased touch wanting[21] in patients. For example, loneliness often co-occurs with chronic illnesses[22], which are linked to depressed mood and feelings of anxiety[23]. Touch can be used to counteract this negative development[24,25]. In adults and children, knowing the toucher did not influence health benefits. In contrast, familiarity affected overall health benefits in newborns, with parental touch being more beneficial than touch applied by medical staff. Previous studies have suggested that early skin-to-skin contact and exposure to maternal odour is critical for a newborn's ability to adapt to a new environment[26], supporting the notion that parental care is difficult to substitute in this time period.

With respect to age-related effects, our data further suggest that increasing age was associated with a higher benefit through touch for systolic blood pressure. These findings could potentially be attributed to higher basal blood pressure[27] with increasing age, allowing for a stronger modulation of this parameter. For sex differences, our study provides some evidence that there are differences between women and men with respect to health benefits of touch. Overall, research on sex differences in touch processing is relatively sparse (but see refs. 28,29). Our results suggest that buffering effects against physiological stress are stronger in women. This is in line with increased buffering effects of hugs in women compared with men[30]. The female-biased primary research in adults, however, begs for more research in men or non-binary individuals. Unfortunately, our study could not dive deeper into this topic as health benefits broken down by sex or gender were almost never provided. Recent research has demonstrated that sensory pleasantness is affected by sex and that this also interacts with

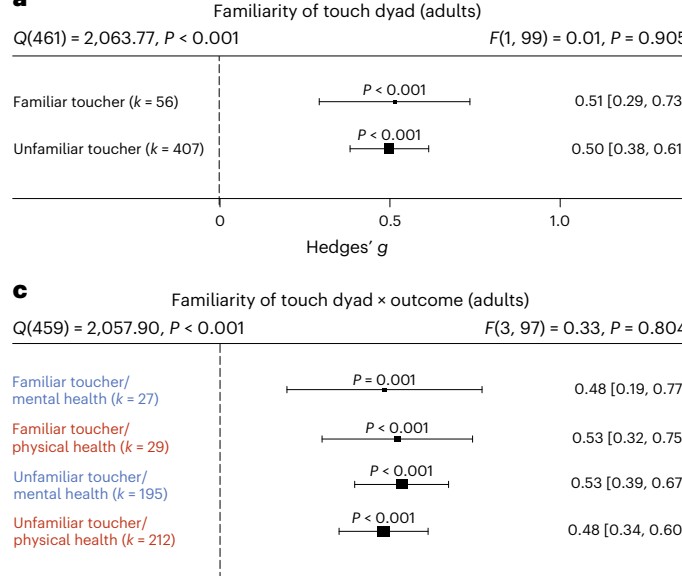

**Fig. 7 | Comparing health benefits for familiar versus unfamiliar touchers.** **a**, Health benefits for being touched by a familiar (for example, partner, family member or friend) versus unfamiliar toucher (health care professional) across 463 in part dependent effect sizes from 83 studies and 101 cohorts. **b**, The same as **a** but for newborn cohorts across 171 in part dependent effect sizes from 51 studies and 62 cohorts. **c**, The same as **a** but separating mental versus physical health benefits across 463 in part dependent effect sizes from 83 studies and 101 cohorts. **d**, The same as **b** but separating mental versus physical health benefits across 169 in part dependent effect sizes from 51 studies and 62 cohorts. Numbers on the right represent the mean effect with its 95% CI in square brackets

and the significance level estimating the likelihood that the effect is equal to zero. Overall effects of moderator impact were assessed via an *F* test, and post hoc comparisons were done using *t* tests (two-sided test). The *F* value in the top right represents a test of the hypothesis that all effects within the subpanel are equal. The *Q* statistic represents the heterogeneity. *P* values of post hoc tests are depicted whenever significant. *P* values above the horizontal whiskers indicate whether an effect differed significantly from a zero effect. Vertical lines indicate significant post hoc tests between moderator levels. *P* values were not corrected for multiple comparisons. Physical outcomes are marked in red. Mental outcomes are marked in blue.

the familiarity of the other person in the touching dyad[29,31]. In general, contextual factors such as sex and gender or the relationship of the touching dyad, differences in cultural background or internal states such as stress have been demonstrated to be highly influential in the perception of affective touch and are thus relevant to maximizing the pleasantness and ultimately the health benefits of touch interactions[32–34]. As a positive personal relationship within the touching dyad is paramount to induce positive health effects, future research applying robot touch to promote well-being should therefore not only explore synthetic skin options but also focus on improving robots as social agents that form a close relationship with the person receiving the touch[35].

As part of the systematic review, we also assessed the effects of touch interventions in non-human animals. Mimicking the results of the meta-analysis in humans, beneficial effects of touch in animals were comparably strong for mental health-like and physical health outcomes. This may inform interventions to promote animal welfare in the context of animal experiments[36], farming[37] and pets[38]. While most studies investigated effects in rodents, which are mostly used as laboratory animals, these results probably transfer to livestock and common pets as well. Indeed, touch was beneficial in lambs, fish and cats[39–41]. The positive impact of human touch in rodents also allows for future mechanistic studies in animal models to investigate how interventions such as tickling or stroking modulate hormonal and neuronal responses to touch in the brain. Furthermore, the commonly proposed oxytocin hypothesis can be causally investigated in these animal models through, for example, optogenetic or chemogenetic techniques[42]. We believe that such translational approaches will further help in optimizing future interventions in humans by uncovering the underlying mechanisms and brain circuits involved in touch.

Our results offer many promising avenues to improve future touch interventions, but they also need to be discussed in light of their limitations. While the majority of findings showed robust health benefits of touch interventions across moderators when compared with a null effect, post hoc tests of, for example, familiarity effects in newborns or mental health benefit differences between human and object touch only barely reached significance. Since we computed a large number of statistical tests in the present study, there is a risk that these results are false positives. We hope that researchers in this field are stimulated by these intriguing results and target these questions by primary research through controlled experimental designs within a well-powered study. Furthermore, the presence of small-study bias in both meta-analyses is indicative that the effect size estimates presented here might be overestimated as null results are often unpublished. We want to stress however that this bias is probably reduced by the multivariate reporting of primary studies. Most studies that reported on multiple health outcomes only showed significant findings for one or two among many. Thus, the multivariate nature of primary research in this field allowed us to include many non-significant findings in the present study. Another limitation pertains to the fact that we only included articles in languages mostly spoken in Western countries. As a large body of evidence comes from Asian countries, it could be that primary research was published in languages other than specified in the inclusion criteria. Thus, despite the large and inclusive nature of our study, some studies could have been missed regardless. Another factor that could not be accounted for in our meta-analysis was that an important prerequisite for touch to be beneficial is its perceived pleasantness. The level of pleasantness associated with being touched is modulated by several parameters[34] including cultural acceptability[43], perceived humanness[44] or a need for touch[45], which could explain the

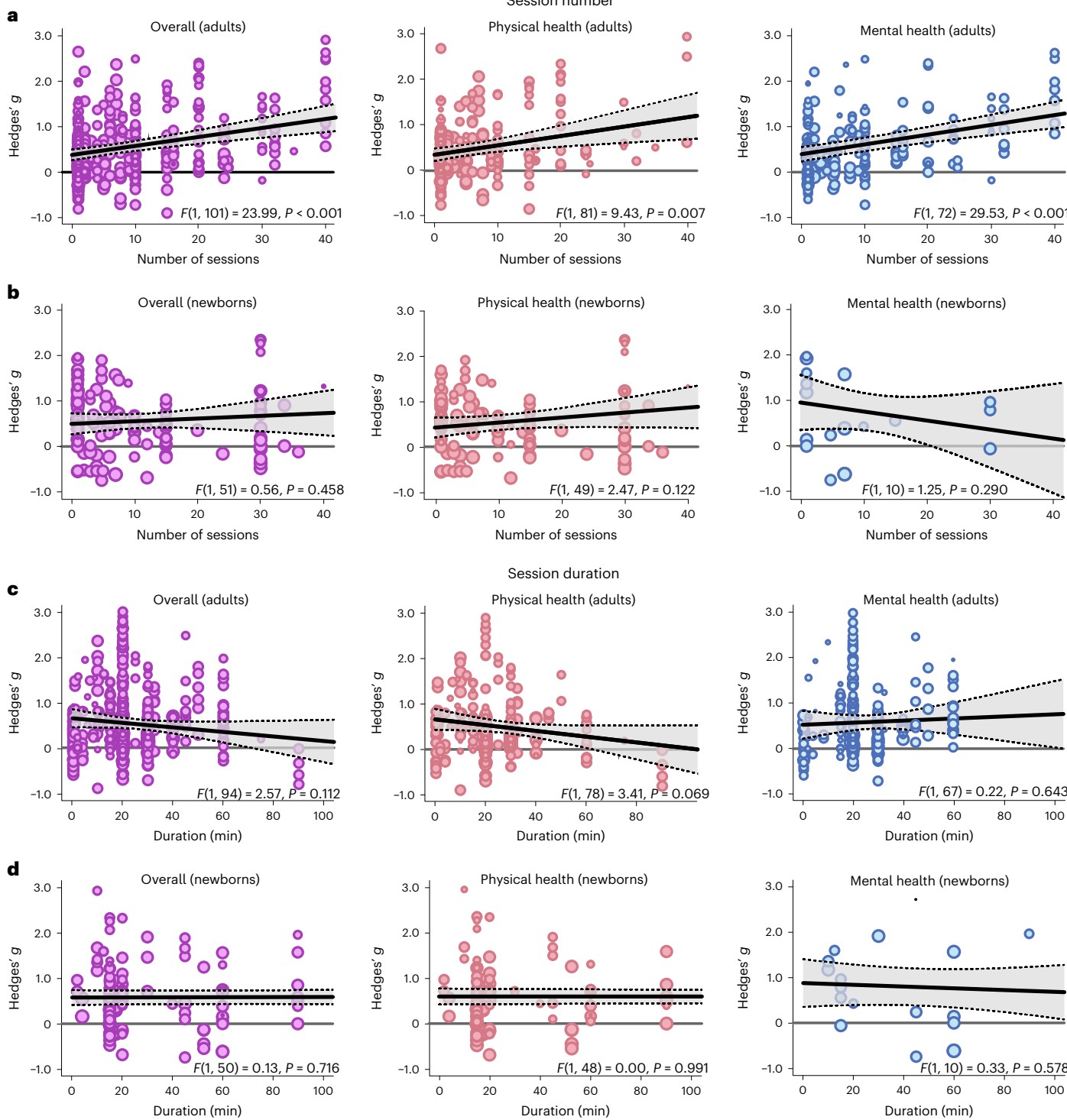

**Fig. 8 | Effect of the number of sessions and their duration on health outcomes. a**, Meta-regression analysis examining the association between the number of sessions applied and the effect size in adults, either on overall health benefits (left, 469 in part dependent effect sizes from 85 studies and 103 cohorts) or for physical (middle, 245 in part dependent effect sizes from 69 studies and 83 cohorts) or mental benefits (right, 224 in part dependent effect sizes from 60 studies and 74 cohorts) separately. **b**, The same as **a** for newborns (overall: 150 in part dependent effect sizes from 46 studies and 53 cohorts; physical health: 127 in part dependent effect sizes from 44 studies and 51 cohorts; mental health: 21 in part dependent effect sizes from 11 studies and 12 cohorts). **c**,**d** the same as **a** (**c**) and **b** (**d**) but for the duration of the individual sessions. For adults, 449 in part dependent effect sizes across 80 studies and 96 cohorts were included in the overall analysis. The analysis of physical health benefits included 240 in

part dependent effect sizes across 67 studies and 80 cohorts, and the analysis of mental health benefits included 209 in part dependent effect sizes from 56 studies and 69 cohorts. For newborns, 145 in part dependent effect sizes across 45 studies and 52 cohorts were included in the overall analysis. The analysis of physical health benefits included 122 in part dependent effect sizes across 43 studies and 50 cohorts, and the analysis of mental health benefits included 21 in part dependent effect sizes from 11 studies and 12 cohorts. Each dot represents an effect size. Its size indicates the precision of the study (larger indicates better). Overall effects of moderator impact were assessed via an $F$ test (two-sided test). The $P$ values in each panel represent the result of a regression analysis testing the hypothesis that the slope of the relationship is equal to zero. $P$ values are not corrected for multiple testing. The shaded area around the regression line represents the 95% CI.

observed differences for certain moderators, such as human–human versus robot–human interaction. Moreover, the fact that secondary categorical moderators could not be investigated with respect to specific health outcomes, owing to the lack of data points, limits the specificity of our conclusions in this regard. It thus remains unclear whether, for example, a decreased mental health benefit in the absence of skin-to-skin contact is linked mostly to decreased anxiolytic effects, changes in positive/negative affect or something else. Since these health outcomes are however highly correlated[46], it is likely that such effects are driven by multiple health outcomes. Similarly, it is important to note that our conclusions mainly refer to outcomes measured close to the touch intervention as we did not include long-term outcomes. Finally, it needs to be noted that blinding towards the experimental condition is essentially impossible in touch interventions. Although we compared the touch intervention with other interventions, such as relaxation therapy, as control whenever possible, contributions of placebo effects cannot be ruled out.

In conclusion, we show clear evidence that touch interventions are beneficial across a large number of both physical and mental health outcomes, for both healthy and clinical cohorts, and for all ages. These benefits, while influenced in their magnitude by study cohorts and intervention characteristics, were robustly present, promoting the conclusion that touch interventions can be systematically employed across the population to preserve and improve our health.

## Methods

### Open science practices

All data and code are accessible in the corresponding OSF project[12]. The systematic review was registered on PROSPERO (CRD42022304281) before the start of data collection. We deviated from the pre-registered plan as follows:

Deviation 1: During our initial screening for the systematic review, we were confronted with a large number of potential health outcomes to look at. This observation of multivariate outcomes led us to register an amendment during data collection (but before any effect size or moderator screening). In doing so, we aimed to additionally extract meta-analytic effects for a more quantitative assessment of our review question that can account for multivariate data reporting and dependencies of effects within the same study. Furthermore, as we noted a severe lack of studies with respect to health outcomes for animals during the inclusion assessment for the systematic review, we decided that the meta-analysis would only focus on outcomes that could be meaningfully analysed on the meta-analytic level and therefore only included health outcomes of human participants.

Deviation 2: In the pre-registration, we did not explicitly exclude non-randomized trials. Since an explicit use of non-randomization for group allocation significantly increases the risk of bias, we decided to exclude them a posteriori from data analysis.

Deviation 3: In the pre-registration, we outlined a tertiary moderator level, namely benefits of touch application versus touch reception. This level was ignored since no included study specifically investigated the benefits of touch application by itself.

Deviation 4: In the pre-registration, we suggested using the RoBMA function[47] to provide a Bayesian framework that allows for a more accurate assessment of publication bias beyond small-study bias. Unfortunately, neither multilevel nor multivariate data structures are supported by the RoBMA function, to our knowledge. For this reason, we did not further pursue this analysis, as the hierarchical nature of the data would not be accounted for.

Deviation 5: Beyond the pre-registered inclusion and exclusion criteria, we also excluded dissertations owing to their lack of peer review.

Deviation 6: In the pre-registration, we stated to investigate the impact of sex of the person applying the touch. This moderator was not further analysed, as this information was rarely given and the individuals applying the touch were almost exclusively women (7 males, 24 mixed and 85 females in studies on adults/children; 3 males, 17 mixed and 80 females in studied on newborns).

Deviation 7: The time span of the touch intervention as assessed by subtracting the final day of the intervention from the first day was not investigated further owing to its very high correlation with the number of sessions ($r(461) = 0.81$ in the adult meta-analysis, $r(145) = 0.84$ in the newborn meta-analysis).

### Inclusion and exclusion criteria

To be included in the systematic review, studies had to investigate the relationship between at least one health outcome (physical and/or mental) in humans or animals and a touch intervention, include explicit physical touch by another human, animal or object as part of an intervention and include an experimental and control condition/group that are differentiated by touch alone. Of note, as a result of this selection process, no animal-to-animal touch intervention study was included, as they never featured a proper no-touch control. Human touch was always explicit touch by a human (that is, no brushes or other tools), either with or without skin-to-skin contact. Regarding the included health outcomes, we aimed to be as broad as possible but excluded parameters such as neurophysiological responses or pleasantness ratings after touch application as they do not reflect health outcomes. All included studies in the meta-analysis and systematic review[48–263] are listed in Supplementary Table 2. All excluded studies are listed in Supplementary Table 3, together with a reason for exclusion. We then applied a two-step process: First, we identified all potential health outcomes and extracted qualitative information on those outcomes (for example, direction of effect). Second, we extracted quantitative information from all possible outcomes (for example, effect sizes). The meta-analysis additionally required a between-subjects design (to clearly distinguish touch from no-touch effects and owing to missing information about the correlation between repeated measurements[264]). Studies that explicitly did not apply a randomized protocol were excluded before further analysis to reduce risk of bias. The full study lists for excluded and included studies can be found in the OSF project[12] in the file 'Study_lists_final_revised.xlsx'. In terms of the time frame, we conducted an open-start search of studies until 2022 and identified studies conducted between 1965 and 2022.

### Data collection

We used Google Scholar, PubMed and Web of Science for our literature search, with no limitations regarding the publication date and using pre-specified search queries (see Supplementary Information for the exact keywords used). All procedures were in accordance with the updated Preferred Reporting Items for Systematic Reviews and Meta-Analyses guidelines[265]. Articles were assessed in French, Dutch, German or English. The above databases were searched from 2 December 2021 until 1 October 2022. Two independent coders evaluated each paper against the inclusion and exclusion criteria. Inconsistencies between coders were checked and resolved by J.P. and H.H. Studies excluded/included for the review and meta-analysis can be found on the OSF project.

### Search queries

We used the following keywords to search the chosen databases. Agents (human versus animal versus object versus robot) and touch outcome (physical versus mental) were searched separately together with keywords searching for touch.

1. TOUCH: Touch OR Social OR Affective OR Contact OR Tactile interaction OR Hug OR Massage OR Embrace OR Kiss OR Cradling OR Stroking OR Haptic interaction OR tickling
2. AGENT: Object OR Robot OR human OR animal OR rodent OR primate

3. MENTAL OUTCOME: Health OR mood OR Depression OR Loneliness OR happiness OR life satisfaction OR Mental Disorder OR well-being OR welfare OR dementia OR psychological OR psychiatric OR anxiety OR Distress
4. PHYSICAL OUTCOME: Health OR Stress OR Pain OR cardiovascular health OR infection risk OR immune response OR blood pressure OR heart rate

### Data extraction and preparation

Data extraction began on 10 October 2022 and was concluded on 25 February 2023. J.P. and H.H. oversaw the data collection process, and checked and resolved all inconsistencies between coders.

Health benefits of touch were always coded by positive summary effects, whereas adverse health effects of touch were represented by negative summary effects. If multiple time points were measured for the same outcome on the same day after a single touch intervention, we extracted the peak effect size (in either the positive or negative direction). If the touch intervention occurred multiple times and health outcomes were assessed for each time point, we extracted data points separately. However, we only extracted immediate effects, as long-term effects not controlled through the experimental conditions could be due to influences other than the initial touch intervention. Measurements assessing long-term effects without explicit touch sessions in the breaks were excluded for the same reason. Common control groups for touch interventions comprised active (for example, relaxation therapy) as well as passive control groups (for example, standard medical care). In the case of multiple control groups, we always contrasted the touch group to the group that most closely matched the touch condition (for example, relaxation therapy was preferred over standard medical care). We extracted information from all moderators listed in the pre-registration (Supplementary Table 4). A list of included and excluded health outcomes is presented in Supplementary Table 5. Authors of studies with possible effects but missing information to calculate those effects were contacted via email and asked to provide the missing data (response rate 35.7%).

After finalizing the list of included studies for the systematic review, we added columns for moderators and the coding schema for our meta-analysis per our updated registration. Then, each study was assessed for its eligibility in the meta-analysis by two independent coders (J.P., H.H., K.F. or F.M.). To this end, all coders followed an a priori specified procedure: First, the PDF was skimmed for possible effects to extract, and the study was excluded if no PDF was available or the study was in a language different from the ones specified in 'Data collection'. Effects from studies that met the inclusion criteria were extracted from all studies listing descriptive values or statistical parameters to calculate effect sizes. A website[266] was used to convert descriptive and statistical values available in the included studies (means and standard deviations/standard errors/confidence intervals, sample sizes, $F$ values, $t$ values, $t$ test $P$ values or frequencies) into Cohen's $d$, which were then converted in Hedges' $g$. If only $P$ value thresholds were reported (for example, $P < 0.01$), we used this, most conservative, value as the $P$ value to calculate the effect size (for example, $P = 0.01$). If only the total sample size was given but that number was even and the participants were randomly assigned to each group, we assumed equal sample sizes for each group. If delta change scores (for example, pre- to post-touch intervention) were reported, we used those over post-touch only scores. In case frequencies were 0 when frequency tables were used to determine effect sizes, we used a value of 0.5 as a substitute to calculate the effect (the default setting in the 'metafor' function[267]). From these data, Hedges' $g$ and its variance could be derived. Effect sizes were always computed between the experimental and the control group.

### Statistical analysis and risk of bias assessment

Owing to the lack of identified studies, health benefits to animals were not included as part of the statistical analysis. One meta-analysis was performed for adults, adolescents and children, as outcomes were highly comparable. We refer to this meta-analysis as the adult meta-analysis, as children/adolescent cohorts were only targeted in a minority of studies. A separate meta-analysis was performed for newborns, as their health outcomes differed substantially from any other age group.

Data were analysed using R (version 4.2.2) with the 'rma.mv' function from the 'metafor' package[267] in a multistep, multivariate and multilevel fashion.

We calculated an overall effect of touch interventions across all studies, cohorts and health outcomes. To account for the hierarchical structure of the data, we used a multilevel structure with random effects at the study, cohort and effects level. Furthermore, we calculated the variance–covariance matrix of all data points to account for the dependencies of measured effects within each individual cohort and study. The variance–covariance matrix was calculated by default with an assumed correlation of effect sizes within each cohort of $\rho = 0.6$. As $\rho$ needed to be assumed, sensitivity analyses for all computed effect estimates were conducted using correlations between effects of 0, 0.2, 0.4 and 0.8. The results of these sensitivity analyses can be found in ref. 12. No conclusion drawn in the present manuscript was altered by changing the level of $\rho$. The sensitivity analyses, however, showed that higher assumed correlations lead to more conservative effect size estimates (see Supplementary Figs. 19 and 20 for the adult and newborn meta-analyses, respectively), reducing the type I error risk in general[268]. In addition to these procedures, we used robust variance estimation with cluster-robust inference at the cohort level. This step is recommended to more accurately determine the confidence intervals in complex multivariate models[269]. The data distribution was assumed to be normal, but this was not formally tested.

To determine whether individual effects had a strong influence on our results, we calculated Cook's distance $D$. Here, a threshold of $D > 0.5$ was used to qualify a study as influential[270]. Heterogeneity in the present study was assessed using Cochran's $Q$, which determines whether the extracted effect sizes estimate a common population effect size. Although the $Q$ statistic in the 'rma.mv' function accounts for the hierarchical nature of the data, we also quantified the heterogeneity estimator $\sigma^2$ for each random-effects level to provide a comprehensive overview of heterogeneity indicators. These indicators for all models can be found on the OSF project[12] in the Table 'Model estimates'. To assess small study bias, we visually inspected the funnel plot and used the standard error as a moderator in the overarching meta-analyses.

Before any sub-group analysis, the overall effect size was used as input for power calculations. While such post hoc power calculations might be limited, we believe that a minimum number of effects to be included in subgroup analyses was necessary to allow for meaningful conclusions. Such medium effect sizes would also probably be the minimum effect sizes of interest for researchers as well as clinical practitioners. Power calculation for random-effects models further requires a sample size for each individual effect as well as an approximation of the expected heterogeneity between studies. For the sample size input, we used the median sample size in each of our studies. For heterogeneity, we assumed a value between medium and high levels of heterogeneity ($I^2 = 62.5\%$[271]), as moderator analyses typically aim at reducing heterogeneity overall. Subgroups were only further investigated if the number of observed effects achieved ~80% power under these circumstances, to allow for a more robust interpretation of the observed effects (see Supplementary Figs. 5 and 6 for the adult and newborn meta-analysis, respectively). In a next step, we investigated all pre-registered moderators for which sufficient power was detected. We first looked at our primary moderators (mental versus physical health) and how the effect sizes systematically varied as a function of our secondary moderators (for example, human–human or human–object touch, duration, skin-to-skin presence, etc.). We always included

random slopes to allow for our moderators to vary with the random effects at our clustering variable, which is recommended in multilevel models to reduce false positives[272]. All statistical tests were performed two-sided. Significance of moderators was determined using omnibus $F$ tests. Effect size differences between moderator levels and their confidence intervals were assessed via $t$ tests.

Post hoc $t$ tests were performed comparing mental and physical health benefits within each interacting moderator (for example, mental versus physical health benefits in cancer patients) and mental or physical health benefits across levels of the interacting moderator (for example, mental health benefits in cancer versus pain patients). The post hoc tests were not pre-registered. Data were visualized using forest plots and orchard plots[273] for categorical moderators and scatter plots for continuous moderators.

For a broad overview of prior work and their biases, risk of bias was assessed for all studies included in both meta-analyses and the systematic review. We assessed the risk of bias for the following parameters:

(1) Bias from randomization, including whether a randomization procedure was performed, whether it was a between- or within-participant design and whether there were any baseline differences for demographic or dependent variables.
(2) Sequence bias resulting from a lack of counterbalancing in within-subject designs.
(3) Performance bias resulting from the participants or experiments not being blinded to the experimental conditions.
(4) Attrition bias resulting from different dropout rates between experimental groups.

Note that four studies in the adult meta-analysis did not explicitly mention randomization as part of their protocol. However, since these studies never showed any baseline differences in all relevant variables (see 'Risk of Bias' table on the OSF project), we assumed that randomization was performed but not mentioned. Sequence bias was of no concern for studies for the meta-analysis since cross-over designs were excluded. It was, however, assessed for studies within the scope of the systematic review. Importantly, performance bias was always high in the adult/children meta-analysis, as blinding of the participants and experimenters to the experimental conditions was not possible owing to the nature of the intervention (touch versus no touch). For studies with newborns and animals, we assessed the performance bias as medium since neither newborns or animals are likely to be aware of being part of an experiment or specific group. An overview of the results is presented in Supplementary Fig. 21, and the precise assessment for each study can be found on the OSF project[12] in the 'Risk of Bias' table.

### Reporting summary
Further information on research design is available in the Nature Portfolio Reporting Summary linked to this article.

## Data availability
All data are available via *Open Science Framework* at https://doi.org/10.17605/OSF.IO/C8RVW (ref. 12). Source data are provided with this paper.

## Code availability
All code is available via *Open Science Framework* at https://doi.org/10.17605/OSF.IO/C8RVW (ref. 12).

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

## Acknowledgements

We thank A. Frick and E. Chris for supporting the initial literature search and coding. We also thank A. Dreisoerner, T. Field, S. Koole, C. Kuhn, M. Henricson, L. Frey Law, J. Fraser, M. Cumella Reddan, and J. Stringer, who kindly responded to our data requests and provided additional information or data with respect to single studies. J.P. was supported by the German National Academy of Sciences Leopoldina (LPDS 2021-05). H.H. was supported by the Marietta-Blau scholarship of the Austrian Agency for Education and Internationalisation (OeAD) and the Deutsche Forschungsgemeinschaft (DFG, German Research Foundation, project ID 422744262 – TRR 289). C.K. received funding from OCENW.XL21.XL21.069 and V.G. from the European Research Council (ERC) under European Union's Horizon 2020 research and innovation programme, grant 'HelpUS' (758703) and from the Dutch Research Council (NWO) grant OCENW.XL21.XL21.069. The funders had no role in study design, data collection and analysis, decision to publish or preparation of the manuscript.

## Author contributions

J.P. contributed to conceptualization, methodology, formal analysis, investigation, data curation, writing the original draft, review and editing, visualization, supervision and project administration. HH contributed to conceptualization, methodology, formal analysis, investigation, data curation, writing the original draft, review and editing, visualization, supervision and project administration. K.F. contributed to investigation, data curation, and review and editing. C.K. and V.G. contributed to conceptualization, and review and editing. F.M. contributed to conceptualization, methodology, formal analysis, investigation, writing the original draft, and review and editing.

## Funding

## Competing interests

The authors declare no competing interests.

## Additional information

**Correspondence and requests for materials** should be addressed to Julian Packheiser.

**Julian Packheiser** ⓘ [1,2,5] ✉, **Helena Hartmann**[2,3,4,5], **Kelly Fredriksen**[2], **Valeria Gazzola** ⓘ [2], **Christian Keysers** ⓘ [2] & **Frédéric Michon** ⓘ [2]

[1]Present address: Social Neuroscience, Faculty of Medicine, Ruhr University Bochum, Bochum, Germany. [2]Social Brain Lab, Netherlands Institute for Neuroscience, Royal Netherlands Academy of Art and Sciences, Amsterdam, the Netherlands. [3]Center for Translational and Behavioral Neuroscience, University Hospital Essen, Essen, Germany. [4]Clinical Neurosciences, Department for Neurology, University Hospital Essen, Essen, Germany. [5]These authors contributed equally: Julian Packheiser, Helena Hartmann. ✉e-mail: julian.packheiser@rub.de

# Reporting Summary

## Statistics

For all statistical analyses, confirm that the following items are present in the figure legend, table legend, main text, or Methods section.

| n/a | Confirmed | |
|---|---|---|
| ☐ | ☒ | The exact sample size (*n*) for each experimental group/condition, given as a discrete number and unit of measurement |
| ☐ | ☒ | A statement on whether measurements were taken from distinct samples or whether the same sample was measured repeatedly |
| ☐ | ☒ | The statistical test(s) used AND whether they are one- or two-sided *Only common tests should be described solely by name; describe more complex techniques in the Methods section.* |
| ☐ | ☒ | A description of all covariates tested |
| ☐ | ☒ | A description of any assumptions or corrections, such as tests of normality and adjustment for multiple comparisons |
| ☐ | ☒ | A full description of the statistical parameters including central tendency (e.g. means) or other basic estimates (e.g. regression coefficient) AND variation (e.g. standard deviation) or associated estimates of uncertainty (e.g. confidence intervals) |
| ☐ | ☒ | For null hypothesis testing, the test statistic (e.g. *F*, *t*, *r*) with confidence intervals, effect sizes, degrees of freedom and *P* value noted *Give P values as exact values whenever suitable.* |
| ☒ | ☐ | For Bayesian analysis, information on the choice of priors and Markov chain Monte Carlo settings |
| ☐ | ☒ | For hierarchical and complex designs, identification of the appropriate level for tests and full reporting of outcomes |
| ☐ | ☒ | Estimates of effect sizes (e.g. Cohen's *d*, Pearson's *r*), indicating how they were calculated |

*Our web collection on statistics for biologists contains articles on many of the points above.*

## Software and code

Policy information about availability of computer code

| Data collection | Data collection was performed from primary studies through a detailed literature search. Effect sizes were collected for each health outcome from each study and a detailed description of the collection process is outlined in the manuscript. Google spreadsheet was used for data collection. |
|---|---|
| Data analysis | Data was analyzed and visualized in R (v4.2.2.) and RStudio (v2023.03.0) using the metafor (v4.4-0), metameta (v.0.2), orchaRd (v.2.1), ggplot2 (v.3.4.4), sandwich (v.3.1-0) and metapower (v.0.2.2) packages. For effect size extraction, we used the website: https://www.campbellcollaboration.org/escalc/html/EffectSizeCalculator-SMD4.php. Data and custom code are fully available to reproduce the analyses under the following link: https://osf.io/c8rvw/. |

For manuscripts utilizing custom algorithms or software that are central to the research but not yet described in published literature, software must be made available to editors and reviewers. We strongly encourage code deposition in a community repository (e.g. GitHub). See the Nature Portfolio guidelines for submitting code & software for further information.

## Data

Policy information about availability of data

All manuscripts must include a data availability statement. This statement should provide the following information, where applicable:

- Accession codes, unique identifiers, or web links for publicly available datasets
- A description of any restrictions on data availability
- For clinical datasets or third party data, please ensure that the statement adheres to our policy

> Data are fully available in the Open Science Framework under the following link: https://osf.io/c8rvw. Data was collected from the following publicly available literature databases: Google Scholar, PubMed and Web of Science. There are no restrictions regarding data availability.

## Research involving human participants, their data, or biological material

Policy information about studies with human participants or human data. See also policy information about sex, gender (identity/presentation), and sexual orientation and race, ethnicity and racism.

| | |
|---|---|
| Reporting on sex and gender | In the present study, we used sex as an analysis factor since most studies reported sex in their studies. In newborns especially, sex is likely to be the variable of interest as a social gender has not yet developed. |
| Reporting on race, ethnicity, or other socially relevant groupings | We used the study location as a proxy for cultural background as moderator in our study. Factors such as race or ethnicity were never reported broken down in such a fashion that a moderation analysis would have been possible. |
| Population characteristics | All relevant characteristics of each individual sample in the meta-analysis have been extracted and used as moderator in the present meta-analysis. |
| Recruitment | No recruitment was part of the study. |
| Ethics oversight | The present meta-analysis did not require ethical approval. |

Note that full information on the approval of the study protocol must also be provided in the manuscript.

# Field-specific reporting

Please select the one below that is the best fit for your research. If you are not sure, read the appropriate sections before making your selection.

☐ Life sciences ☒ Behavioural & social sciences ☐ Ecological, evolutionary & environmental sciences

For a reference copy of the document with all sections, see nature.com/documents/nr-reporting-summary-flat.pdf

# Life sciences study design

All studies must disclose on these points even when the disclosure is negative.

| | |
|---|---|
| Sample size | *Describe how sample size was determined, detailing any statistical methods used to predetermine sample size OR if no sample-size calculation was performed, describe how sample sizes were chosen and provide a rationale for why these sample sizes are sufficient.* |
| Data exclusions | *Describe any data exclusions. If no data were excluded from the analyses, state so OR if data were excluded, describe the exclusions and the rationale behind them, indicating whether exclusion criteria were pre-established.* |
| Replication | *Describe the measures taken to verify the reproducibility of the experimental findings. If all attempts at replication were successful, confirm this OR if there are any findings that were not replicated or cannot be reproduced, note this and describe why.* |
| Randomization | *Describe how samples/organisms/participants were allocated into experimental groups. If allocation was not random, describe how covariates were controlled OR if this is not relevant to your study, explain why.* |
| Blinding | *Describe whether the investigators were blinded to group allocation during data collection and/or analysis. If blinding was not possible, describe why OR explain why blinding was not relevant to your study.* |

# Behavioural & social sciences study design

All studies must disclose on these points even when the disclosure is negative.

| | |
|---|---|
| Study description | The study is constituted of two quantitative meta-analyses as well as a more qualitative systematic review about the efficacy of touch interventions and the moderating factors that influence its efficacy. |

| | |
|---|---|
| Research sample | In total, 166 different cohorts were tested across both meta-analyses. These cohorts had a large number of different backgrounds and varied greatly with respect to demographic variables. Thus, the overall effect reported in this paper is highly representative. Heterogeneity was investigated through moderation analyses. Relevant demographic information regarding for example sex ratios or mean ages are available in the OSF file "Data Final.xlsx" (Sheets: AdultsChildren Final datasheet/Newborns Final datasheet) for each individual primary study if this information was available. The rational to include a highly diverse sample with different demographic backgrounds was to be inclusive and representative while being able to identify moderating roles of such variables. As this study constitutes a meta-analysis and systematic review, previously published data was used for further analysis. The source of the data were original publications as searched via Google Scholar, PubMed and Web of Science. |
| Sampling strategy | We used Google Scholar, PubMed and Web of Science for our literature search. The following search terms were used to identify all relevant studies: Agents (human vs. animal vs. object vs. robot) and touch outcome (physical vs. mental) were searched separately together with keywords searching for touch.<br><br>1. TOUCH: Touch OR Social OR Affective OR Contact OR Tactile interaction OR Hug OR Massage OR Embrace OR Kiss OR Cradling OR Stroking OR Haptic interaction OR tickling<br>2. AGENT: Object OR Robot OR human OR animal OR rodent OR primate<br>3. MENTAL OUTCOME: Health OR mood OR Depression OR Loneliness OR happiness OR life satisfaction OR Mental Disorder OR well-being OR welfare OR dementia OR psychological OR psychiatric OR anxiety OR Distress<br>4. PHYSICAL OUTCOME: Health OR Stress OR Pain OR cardiovascular health OR infection risk OR immune response OR blood pressure OR heart rate<br><br>As we searched reference lists from studies found through these search terms, we used a snowball sampling technique. |
| Data collection | We used a Google spreadsheet for data collection for both the literature search and formal data extraction. Effect size calculation was done both via a dedicated website (https://www.campbellcollaboration.org/escalc/html/EffectSizeCalculator-SMD4.php) and within the metafor function. Researchers were not blinded to the hypotheses during data collection. Blinding to experimental conditions does not apply as no experiments were conducted in this study. |
| Timing | The databases were searched from 2nd of December 2021 until the 01th of October 2022. Data extraction began on the 10th of October 2022 and was concluded on the 25th of February 2023. |
| Data exclusions | Exclusion criteria were established and detailed in the pre-registration prior to study onset. All study exclusions are listed in detail in the flowchart (Figure 1). Overall, 750 records were excluded. |
| Non-participation | No participants were involved in the present study as it constitutes a meta-analysis of existing data. |
| Randomization | Randomization was assessed to identify risk of bias. Explicit non-randomization was an exclusion criterion due to heightened risk of bias. |

# Ecological, evolutionary & environmental sciences study design

All studies must disclose on these points even when the disclosure is negative.

| | |
|---|---|
| Study description | *Briefly describe the study. For quantitative data include treatment factors and interactions, design structure (e.g. factorial, nested, hierarchical), nature and number of experimental units and replicates.* |
| Research sample | *Describe the research sample (e.g. a group of tagged Passer domesticus, all Stenocereus thurberi within Organ Pipe Cactus National Monument), and provide a rationale for the sample choice. When relevant, describe the organism taxa, source, sex, age range and any manipulations. State what population the sample is meant to represent when applicable. For studies involving existing datasets, describe the data and its source.* |
| Sampling strategy | *Note the sampling procedure. Describe the statistical methods that were used to predetermine sample size OR if no sample-size calculation was performed, describe how sample sizes were chosen and provide a rationale for why these sample sizes are sufficient.* |
| Data collection | *Describe the data collection procedure, including who recorded the data and how.* |
| Timing and spatial scale | *Indicate the start and stop dates of data collection, noting the frequency and periodicity of sampling and providing a rationale for these choices. If there is a gap between collection periods, state the dates for each sample cohort. Specify the spatial scale from which the data are taken* |
| Data exclusions | *If no data were excluded from the analyses, state so OR if data were excluded, describe the exclusions and the rationale behind them, indicating whether exclusion criteria were pre-established.* |
| Reproducibility | *Describe the measures taken to verify the reproducibility of experimental findings. For each experiment, note whether any attempts to repeat the experiment failed OR state that all attempts to repeat the experiment were successful.* |
| Randomization | *Describe how samples/organisms/participants were allocated into groups. If allocation was not random, describe how covariates were controlled. If this is not relevant to your study, explain why.* |
| Blinding | *Describe the extent of blinding used during data acquisition and analysis. If blinding was not possible, describe why OR explain why blinding was not relevant to your study.* |

| | Did the study involve field work? | ☐ Yes | ☐ No |
|---|---|---|---|

## Field work, collection and transport

| | |
|---|---|
| Field conditions | *Describe the study conditions for field work, providing relevant parameters (e.g. temperature, rainfall).* |
| Location | *State the location of the sampling or experiment, providing relevant parameters (e.g. latitude and longitude, elevation, water depth).* |
| Access & import/export | *Describe the efforts you have made to access habitats and to collect and import/export your samples in a responsible manner and in compliance with local, national and international laws, noting any permits that were obtained (give the name of the issuing authority, the date of issue, and any identifying information).* |
| Disturbance | *Describe any disturbance caused by the study and how it was minimized.* |

# Reporting for specific materials, systems and methods

We require information from authors about some types of materials, experimental systems and methods used in many studies. Here, indicate whether each material, system or method listed is relevant to your study. If you are not sure if a list item applies to your research, read the appropriate section before selecting a response.

### Materials & experimental systems

| n/a | Involved in the study |
|---|---|
| ☒ | ☐ Antibodies |
| ☒ | ☐ Eukaryotic cell lines |
| ☒ | ☐ Palaeontology and archaeology |
| ☒ | ☐ Animals and other organisms |
| ☒ | ☐ Clinical data |
| ☒ | ☐ Dual use research of concern |
| ☒ | ☐ Plants |

### Methods

| n/a | Involved in the study |
|---|---|
| ☒ | ☐ ChIP-seq |
| ☒ | ☐ Flow cytometry |
| ☒ | ☐ MRI-based neuroimaging |

## Antibodies

| | |
|---|---|
| Antibodies used | *Describe all antibodies used in the study; as applicable, provide supplier name, catalog number, clone name, and lot number.* |
| Validation | *Describe the validation of each primary antibody for the species and application, noting any validation statements on the manufacturer's website, relevant citations, antibody profiles in online databases, or data provided in the manuscript.* |

## Eukaryotic cell lines

Policy information about cell lines and Sex and Gender in Research

| | |
|---|---|
| Cell line source(s) | *State the source of each cell line used and the sex of all primary cell lines and cells derived from human participants or vertebrate models.* |
| Authentication | *Describe the authentication procedures for each cell line used OR declare that none of the cell lines used were authenticated.* |
| Mycoplasma contamination | *Confirm that all cell lines tested negative for mycoplasma contamination OR describe the results of the testing for mycoplasma contamination OR declare that the cell lines were not tested for mycoplasma contamination.* |
| Commonly misidentified lines (See ICLAC register) | *Name any commonly misidentified cell lines used in the study and provide a rationale for their use.* |

## Palaeontology and Archaeology

| | |
|---|---|
| Specimen provenance | *Provide provenance information for specimens and describe permits that were obtained for the work (including the name of the issuing authority, the date of issue, and any identifying information). Permits should encompass collection and, where applicable, export.* |
| Specimen deposition | *Indicate where the specimens have been deposited to permit free access by other researchers.* |
| Dating methods | *If new dates are provided, describe how they were obtained (e.g. collection, storage, sample pretreatment and measurement), where* |

| Dating methods | *they were obtained (i.e. lab name), the calibration program and the protocol for quality assurance OR state that no new dates are provided.* |

☐ Tick this box to confirm that the raw and calibrated dates are available in the paper or in Supplementary Information.

| Ethics oversight | *Identify the organization(s) that approved or provided guidance on the study protocol, OR state that no ethical approval or guidance was required and explain why not.* |

Note that full information on the approval of the study protocol must also be provided in the manuscript.

# Animals and other research organisms

Policy information about studies involving animals; ARRIVE guidelines recommended for reporting animal research, and Sex and Gender in Research

| Laboratory animals | *For laboratory animals, report species, strain and age OR state that the study did not involve laboratory animals.* |

| Wild animals | *Provide details on animals observed in or captured in the field; report species and age where possible. Describe how animals were caught and transported and what happened to captive animals after the study (if killed, explain why and describe method; if released, say where and when) OR state that the study did not involve wild animals.* |

| Reporting on sex | *Indicate if findings apply to only one sex; describe whether sex was considered in study design, methods used for assigning sex. Provide data disaggregated for sex where this information has been collected in the source data as appropriate; provide overall numbers in this Reporting Summary. Please state if this information has not been collected. Report sex-based analyses where performed, justify reasons for lack of sex-based analysis.* |

| Field-collected samples | *For laboratory work with field-collected samples, describe all relevant parameters such as housing, maintenance, temperature, photoperiod and end-of-experiment protocol OR state that the study did not involve samples collected from the field.* |

| Ethics oversight | *Identify the organization(s) that approved or provided guidance on the study protocol, OR state that no ethical approval or guidance was required and explain why not.* |

Note that full information on the approval of the study protocol must also be provided in the manuscript.

# Clinical data

Policy information about clinical studies

All manuscripts should comply with the ICMJE guidelines for publication of clinical research and a completed CONSORT checklist must be included with all submissions.

| Clinical trial registration | *Provide the trial registration number from ClinicalTrials.gov or an equivalent agency.* |

| Study protocol | *Note where the full trial protocol can be accessed OR if not available, explain why.* |

| Data collection | *Describe the settings and locales of data collection, noting the time periods of recruitment and data collection.* |

| Outcomes | *Describe how you pre-defined primary and secondary outcome measures and how you assessed these measures.* |

# Dual use research of concern

Policy information about dual use research of concern

## Hazards

Could the accidental, deliberate or reckless misuse of agents or technologies generated in the work, or the application of information presented in the manuscript, pose a threat to:

No | Yes

☐ ☐ Public health

☐ ☐ National security

☐ ☐ Crops and/or livestock

☐ ☐ Ecosystems

☐ ☐ Any other significant area

## Experiments of concern

Does the work involve any of these experiments of concern:

No | Yes
--- | ---
☐ | ☐ Demonstrate how to render a vaccine ineffective
☐ | ☐ Confer resistance to therapeutically useful antibiotics or antiviral agents
☐ | ☐ Enhance the virulence of a pathogen or render a nonpathogen virulent
☐ | ☐ Increase transmissibility of a pathogen
☐ | ☐ Alter the host range of a pathogen
☐ | ☐ Enable evasion of diagnostic/detection modalities
☐ | ☐ Enable the weaponization of a biological agent or toxin
☐ | ☐ Any other potentially harmful combination of experiments and agents

## Plants

**Seed stocks**
Report on the source of all seed stocks or other plant material used. If applicable, state the seed stock centre and catalogue number. If plant specimens were collected from the field, describe the collection location, date and sampling procedures.

**Novel plant genotypes**
Describe the methods by which all novel plant genotypes were produced. This includes those generated by transgenic approaches, gene editing, chemical/radiation-based mutagenesis and hybridization. For transgenic lines, describe the transformation method, the number of independent lines analyzed and the generation upon which experiments were performed. For gene-edited lines, describe the editor used, the endogenous sequence targeted for editing, the targeting guide RNA sequence (if applicable) and how the editor was applied.

**Authentication**
Describe any authentication procedures for each seed stock used or novel genotype generated. Describe any experiments used to assess the effect of a mutation and, where applicable, how potential secondary effects (e.g. second site T-DNA insertions, mosiacism, off-target gene editing) were examined.

## ChIP-seq

### Data deposition

☐ Confirm that both raw and final processed data have been deposited in a public database such as GEO.

☐ Confirm that you have deposited or provided access to graph files (e.g. BED files) for the called peaks.

**Data access links**
*May remain private before publication.*
For "Initial submission" or "Revised version" documents, provide reviewer access links. For your "Final submission" document, provide a link to the deposited data.

**Files in database submission**
Provide a list of all files available in the database submission.

**Genome browser session**
(e.g. UCSC)
Provide a link to an anonymized genome browser session for "Initial submission" and "Revised version" documents only, to enable peer review. Write "no longer applicable" for "Final submission" documents.

### Methodology

**Replicates**
Describe the experimental replicates, specifying number, type and replicate agreement.

**Sequencing depth**
Describe the sequencing depth for each experiment, providing the total number of reads, uniquely mapped reads, length of reads and whether they were paired- or single-end.

**Antibodies**
Describe the antibodies used for the ChIP-seq experiments; as applicable, provide supplier name, catalog number, clone name, and lot number.

**Peak calling parameters**
Specify the command line program and parameters used for read mapping and peak calling, including the ChIP, control and index files used.

**Data quality**
Describe the methods used to ensure data quality in full detail, including how many peaks are at FDR 5% and above 5-fold enrichment.

**Software**
Describe the software used to collect and analyze the ChIP-seq data. For custom code that has been deposited into a community repository, provide accession details.

# Flow Cytometry

## Plots

Confirm that:

☐ The axis labels state the marker and fluorochrome used (e.g. CD4-FITC).

☐ The axis scales are clearly visible. Include numbers along axes only for bottom left plot of group (a 'group' is an analysis of identical markers).

☐ All plots are contour plots with outliers or pseudocolor plots.

☐ A numerical value for number of cells or percentage (with statistics) is provided.

## Methodology

| | |
|---|---|
| Sample preparation | *Describe the sample preparation, detailing the biological source of the cells and any tissue processing steps used.* |
| Instrument | *Identify the instrument used for data collection, specifying make and model number.* |
| Software | *Describe the software used to collect and analyze the flow cytometry data. For custom code that has been deposited into a community repository, provide accession details.* |
| Cell population abundance | *Describe the abundance of the relevant cell populations within post-sort fractions, providing details on the purity of the samples and how it was determined.* |
| Gating strategy | *Describe the gating strategy used for all relevant experiments, specifying the preliminary FSC/SSC gates of the starting cell population, indicating where boundaries between "positive" and "negative" staining cell populations are defined.* |

☐ Tick this box to confirm that a figure exemplifying the gating strategy is provided in the Supplementary Information.

# Magnetic resonance imaging

## Experimental design

| | |
|---|---|
| Design type | *Indicate task or resting state; event-related or block design.* |
| Design specifications | *Specify the number of blocks, trials or experimental units per session and/or subject, and specify the length of each trial or block (if trials are blocked) and interval between trials.* |
| Behavioral performance measures | *State number and/or type of variables recorded (e.g. correct button press, response time) and what statistics were used to establish that the subjects were performing the task as expected (e.g. mean, range, and/or standard deviation across subjects).* |

## Acquisition

| | |
|---|---|
| Imaging type(s) | *Specify: functional, structural, diffusion, perfusion.* |
| Field strength | *Specify in Tesla* |
| Sequence & imaging parameters | *Specify the pulse sequence type (gradient echo, spin echo, etc.), imaging type (EPI, spiral, etc.), field of view, matrix size, slice thickness, orientation and TE/TR/flip angle.* |
| Area of acquisition | *State whether a whole brain scan was used OR define the area of acquisition, describing how the region was determined.* |

Diffusion MRI     ☐ Used     ☐ Not used

## Preprocessing

| | |
|---|---|
| Preprocessing software | *Provide detail on software version and revision number and on specific parameters (model/functions, brain extraction, segmentation, smoothing kernel size, etc.).* |
| Normalization | *If data were normalized/standardized, describe the approach(es): specify linear or non-linear and define image types used for transformation OR indicate that data were not normalized and explain rationale for lack of normalization.* |
| Normalization template | *Describe the template used for normalization/transformation, specifying subject space or group standardized space (e.g. original Talairach, MNI305, ICBM152) OR indicate that the data were not normalized.* |
| Noise and artifact removal | *Describe your procedure(s) for artifact and structured noise removal, specifying motion parameters, tissue signals and physiological signals (heart rate, respiration).* |

| Volume censoring | *Define your software and/or method and criteria for volume censoring, and state the extent of such censoring.* |
|---|---|

## Statistical modeling & inference

| Model type and settings | *Specify type (mass univariate, multivariate, RSA, predictive, etc.) and describe essential details of the model at the first and second levels (e.g. fixed, random or mixed effects; drift or auto-correlation).* |
|---|---|
| Effect(s) tested | *Define precise effect in terms of the task or stimulus conditions instead of psychological concepts and indicate whether ANOVA or factorial designs were used.* |

Specify type of analysis:   ☐ Whole brain   ☐ ROI-based   ☐ Both

| Statistic type for inference | *Specify voxel-wise or cluster-wise and report all relevant parameters for cluster-wise methods.* |
|---|---|

(See Eklund et al. 2016)

| Correction | *Describe the type of correction and how it is obtained for multiple comparisons (e.g. FWE, FDR, permutation or Monte Carlo).* |
|---|---|

## Models & analysis

| n/a | Involved in the study |
|---|---|
| ☐ | ☐ Functional and/or effective connectivity |
| ☐ | ☐ Graph analysis |
| ☐ | ☐ Multivariate modeling or predictive analysis |

| Functional and/or effective connectivity | *Report the measures of dependence used and the model details (e.g. Pearson correlation, partial correlation, mutual information).* |
|---|---|
| Graph analysis | *Report the dependent variable and connectivity measure, specifying weighted graph or binarized graph, subject- or group-level, and the global and/or node summaries used (e.g. clustering coefficient, efficiency, etc.).* |
| Multivariate modeling and predictive analysis | *Specify independent variables, features extraction and dimension reduction, model, training and evaluation metrics.* |

