## [Peer Review File · Nature Human Behaviour]

Peer Review Information

Journal: Nature Human Behaviour

Manuscript Title: A systematic review and multivariate meta-analysis of the physical and mental health benefits of touch interventions

Corresponding author name(s): Julian Packheiser

Reviewer Comments & Decisions:

Decision Letter, initial version:

18th September 2023

Dear Dr Packheiser,

Thank you once again for your manuscript, entitled "The physical and mental health benefits of touch interventions: A comparative systematic review and multivariate meta-analysis," and for your patience during the peer review process.

Your manuscript has now been evaluated by 3 reviewers, whose comments are included at the end of this letter. Although the reviewers find your work to be of interest, they also raise some important concerns. We are very interested in the possibility of publishing your study in Nature Human Behaviour, but would like to consider your response to these concerns in the form of a revised manuscript before we make a decision on publication.

To guide the scope of the revisions, the editors discuss the referee reports in detail within the team, including with the chief editor, with a view to (1) identifying key priorities that should be addressed in revision and (2) overruling referee requests that are deemed beyond the scope of the current study. We hope that you will find the prioritised set of referee points to be useful when revising your study. Please do not hesitate to get in touch if you would like to discuss these issues further.

1. Our reviewers ask for additional methodological details and examples to be included. Similarly, they request clarification of technical details and choices. Please carefully revise your work in line with the reviewer comments, and explicitly motivate all methodological and technical choices.
2. Reviewer 1 and Reviewer 2 raise questions regarding the way in which small study bias was assessed. Please explicitly motivate your approach, and consider using the bias-reduced linearization adjustment proposed by Bell and McCaffrey (2002), as recommended by Reviewer 1, instead.
3. Reviewer 2 mentions that $\rho=0.6$ is assumed in calculating the variance-covariance matrix. To rule out that this value impacts the results, the reviewer asks that you conduct a sensitivity analysis by using different values and see if the results are robust.
4. Reviewer 1 finds the comparative part of the manuscript to be out of scope of the current meta-analysis and systematic review. While we agree with the reviewer that the comparative part should not be the main focus of the manuscript, we do believe that there is value in these comparisons and would not encourage you to remove them. Instead, we ask you to revise the title of your manuscript, as well as the main text, to reflect that the main focus of your work lies on the effects of human touch interventions on mental and physical health. Please briefly mention comparative research in the introduction, noting already the sparseness.

Finally, your revised manuscript must comply fully with our editorial policies and formatting requirements. Failure to do so will result in your manuscript being returned to you, which will delay

its consideration. To assist you in this process, I have attached a checklist that lists all of our requirements. If you have any questions about any of our policies or formatting, please don't hesitate to contact me.

In sum, we invite you to revise your manuscript taking into account all reviewer and editor comments. We are committed to providing a fair and constructive peer-review process. Do not hesitate to contact us if there are specific requests from the reviewers that you believe are technically impossible or unlikely to yield a meaningful outcome.

We hope to receive your revised manuscript within two months. I would be grateful if you could contact us as soon as possible if you foresee difficulties with meeting this target resubmission date.

- Include a "Response to the editors and reviewers" document detailing, point-by-point, how you addressed each editor and referee comment. If no action was taken to address a point, you must provide a compelling argument. When formatting this document, please respond to each reviewer comment individually, including the full text of the reviewer comment verbatim followed by your response to the individual point. This response will be used by the editors to evaluate your revision and sent back to the reviewers along with the revised manuscript.
- Highlight all changes made to your manuscript or provide us with a version that tracks changes.

[REDACTED]

We look forward to seeing the revised manuscript and thank you for the opportunity to review your work. Please do not hesitate to contact me if you have any questions or would like to discuss these revisions further.

Sincerely,

Samantha Antusch

Samantha Antusch, PhD
Senior Editor
Nature Human Behaviour

Reviewer expertise:

Reviewer #1: touch interventions for physical health and touch interventions for mental health

Reviewer #2: meta-analysis

Reviewer #3: touch interventions for mental health

REVIEWER COMMENTS:

Reviewer #1:
Remarks to the Author:
Overview:

The manuscript presents a multivariate meta-analysis and comparative systematic review on the physical and mental health benefits of touch interventions. The work brings together and analyses 137 studies (166 cohorts, 9617 participants, and 643 effect sizes) examining physical and mental health benefits of various types of touch therapies (e.g., massage, kangaroo care, hugging, handholding) for both clinical groups and healthy subjects. By combining various lines of interventions, which target different subject groups (newborn, children, adult, clinical, and healthy cohorts) and by examining the influence of various moderators, the manuscript makes a significant contribution to the scientific understanding of the general physical and mental health benefits of touch interventions. Moreover, there are many highly intriguing results, which could be used in the development of more effective touch interventions and tailoring of touch interventions for specific target groups and clinical/social contexts. For example, the finding that newborns benefit most from a touch provided by a familiar person whereas adults and children benefit similarly from a touch provided by an unfamiliar and familiar sender is highly intriguing and calls for further examination. Similarly, the finding that a robot/object touch has comparable benefits for adult physical health, but poorer mental health outcomes as compared to human-human touch is likewise a result with high clinical/practical relevance.

Validity and robustness of the interpretations:

Reading the results, examining the figures, and inspecting all the supplementary files did not bring up flaws that would question the validity or robustness of the main results. The only reservations I had related to the subgroup analyses (and post hoc comparisons) resulting in a large number of statistical tests. Many of the significant p-values tended to stay close to the .05 threshold (e.g. familiarity of toucher in newborns: $p = .041$; object vs. human touch on adult mental health: $p = .022$). Running many subgroup analyses and pairwise comparisons while assuming a 5% false positive rate is likely to result in some false positives. Can the authors comment on whether and how they handled the type I error risk in the subgroup analyses?

Significance:

Previous meta-analyses have already demonstrated the health benefits of massage therapy in adults (Moyer et al., 2004; Lee et al., 2015) and kangaroo care in newborns (Boundy et al., 2016). The significance of the current work is that it brings together and enables comparison between these different lines of research while also presenting a larger set of studies with a more extensive set of cohort types, outcomes, and moderators. Therefore, the new meta-analysis provides a more conclusive picture of the health benefits of touch interventions than the previous meta-analyses. Moreover, many of the results may be helpful in developing more effective touch interventions and when designing haptics and mediated touch technologies.

Analytical approach:

The sample size is also feasible for multivariate meta-analysis. The total sample of studies was further divided into two meta-analyses run separately for the child/adult and the newborn studies, which was justifiable due to the systematic difference between the outcome measures in these two subject groups. To demonstrate sufficient power for subgroup analyses, the authors ran power calculations and examined only those secondary moderators, for which a sufficient number of observations could be found. This, I think, was the right decision due to the large number of sub-group analyses and the limited amount of data for some of the sub-groups.

The analyses took into account multilevel data structure, the influence of single studies, heterogeneity of the effects sizes, and small study bias. Moreover, the authors analyzed the risks of bias in randomization, sequence, performance, and attrition. However, the results of these analyses were not elaborated in the discussion. The same is the case with funnel plots, which showed signs of publication bias as there was a scarcity of smaller studies at the bottom of the funnel. Moreover, the formal test for publication bias was significant for the children/adults set and trending for the newborn set. If understood correctly, the small study bias was taken into account in the analysis by setting the variance of the effect sizes as a moderator in the overarching meta-analyses. From the analysis script, one can see that `robust()` -function of the `metafor` -package was used to provide cluster-robust tests and confidence intervals for the meta-analysis model coefficients. The script indicates that as an adjustment method, the cluster-robust estimate of the variance-covariance

matrix was multiplied by the factor $n/(n-p)$ where n refers to the number of clusters and p denotes the total number of model coefficients. Does this method serve as the small-sample adjustment method? Although this adjustment method was applied as a default method by the `robust()` function, it would be good to indicate it in the method section. In the `robust()` function documentation (<https://wviechtb.github.io/metafor/reference/robust.html>), a better small-sample adjustment method is presented utilizing a bias-reduced linearization adjustment proposed by Bell and McCaffrey (2002). Can the authors explain why the default method was used instead of this?

Suggested improvements

A more general thing I wanted to comment on is related to the comparative part. That is, the word comparative is in the title but this aspect is pretty much absent in the introduction, methods, and results. In the Results section, there is a small paragraph briefly describing findings regarding the health outcomes of touch in other species. In the discussion, a whole paragraph is used to discuss these findings and their relevance. Thus, there is a clear imbalance between the sections in regard to the comparative aspect. Already in the pre-registration, the lack of animal studies was acknowledged and it was clear that these studies would not be able to be incorporated into the meta-analysis. I consider the health benefits of touching on non-human animals a separate question and out of the scope of the current review/meta-analysis but how do the authors and especially the editor see this?

In my view, in the discussion, one could have elaborated more on the relation of the current meta-analysis results to the results of the previous more limited meta-analyses. Moreover, one could have elaborated the results in relation to previous research on how sex and gender influence sensory pleasantness and social comfort associated with touch (Schirmer et al., 2022, *Physiology & Behavior*) or how attributing tactile stimulation to a social agent (another human vs. robot vs. virtual agent) could boost the mental health benefits of non-skin-to-skin touch (Huisman et al., 2017, *IEEE Trans. Hapt Ravaja et al., 2017, Scientific Reports*).

Finally, one could have pointed out the risk of alpha inflation in the subgroup analyses as a limitation in the discussion.

Clarity and context

The manuscript was clearly written and gave sufficient context to the reader to ensure accessibility of the text to the wide readership of *Nature Human Behavior*. In the abstract, I would have preferred a less technical style of stating the knowledge gap. That is, instead of referring to the effect size and moderators in the second sentence, one could have spoken about these aspects in more layperson terms (e.g., "what factors make touch interventions beneficial for physical and mental health").

References

I advise going through the list of references carefully because at least Lee et al., 2015 was missing from the list of references.

Conclusion

To conclude, the manuscript presents a large-scale systematic review and meta-analysis of the health benefits of touch interventions with significant contributions to the field. There are a few notions or concerns about some of the secondary moderators being false positives due to alpha inflation. Keeping this risk in mind, the overall results seem robust and significant.

Kind regards,
Dr. Ville Harjunen

Reviewer #2:

Remarks to the Author:

This study conducted a systematic review and a large-scale meta-analysis involving 137 studies with healthy participants and patients to address the benefits of touch on mental and physical health. My research area is meta-analysis, so I will mainly comment on the methodology.

1. Lines 196-198. Hedges' g is the primary effect size. However, it is mentioned that effect sizes based on frequency tables are also included in the meta-analysis. How many studies are in frequency tables? What formulas and assumptions for transforming odds ratios to Hedges' g ?
2. Lines 213-214. $\rho=0.6$ is assumed in calculating the variance-covariance matrix. Since this value may impact the results, it is necessary to conduct a sensitivity analysis by using different values and see if the results are robust.
3. Lines 221-222. The sampling variance of the effect size is used as a moderator in assessing small study bias. Using standard error (SE) is more popular. Can you elaborate on why the sampling variance, not SE, is used?
4. Lines 223-231. Post-hoc power is calculated. It is generally accepted that post-hoc power analysis is not useful. If the observed data is statistically significant, the post-hoc power is large. On the other hand, the post-hoc power is low when the observed data is not statistically significant. I suggest dropping the post-hoc power calculation. If the authors want to keep it, please provide the justifications.
5. There are no heterogeneity estimates in the results and the figures. Is it because heterogeneity estimates are not available in robust variance estimation? If yes, please explain this limitation.
6. In the spreadsheet "AdultsChildren Final datasheet") of the data file "Data final.xlsx," there is a formula (column T) to correct the small bias of d (column S) to Hedges' g . However, there is no formula in the sampling variance (column U). Thus, it is unclear whether the sampling variance in column U is for d or Hedges' g . Please clarify.

Reviewer #3:

Remarks to the Author:

Packheiser & Hartmann et al. present a timely and important meta-analysis of potential health benefits of touch interventions. Across a large body of studies they find medium-sized beneficial effects of touch on both mental and physical health outcome measures, which they analyze with regard to sub-populations or factors whenever statistically appropriate.

This is an interesting and important work summarizing the somewhat hard to compare literature on touch benefits.

I only have minor comments:

Methods: What was the time range of the literature included in the search?

Language of the included articles is unclear. You write French, Dutch, German, English in the text, but the table says some were excluded for not being in English.

How were the specific outcome measures chosen?

For the meta-analyses, only between-subject designs were included, however, in the bias analysis you mention both between- and within-designs. Please clarify. Also regarding the bias, you include not being blinded in the parameters, however, I assume it not to be possible to be blinded to conditions in a touch vs. no touch study.

Where did you draw the line between human and object/robot touch? Only skin-to-skin vs. all other?

What about e.g. handheld tools (brushes) or what about touch that is not skin-to-skin (through clothing or with gloves)?

Results/Discussion: While I appreciate your clean and fact based style, I think the article would benefit from a few examples and descriptors of both the touch interventions and the control conditions, e.g. the most common types used.

What was the standard deviation of age in the meta analysis sample?

Could you include a list of all touched body parts and of all types of touch interventions in the supplement?

Animal studies: are these only touches by humans to the animal or also touch by members of the same species?

Discussion: While it is to be assumed that most scientific peer-reviewed studies are published in English language, the specific field of research of touch benefits on health outcomes might also be

addressed in a larger body of articles in languages other than English, considering that manual practices and massage therapy are actually more common in countries like China and India. This might be a limitation, as these studies were not included. In addition, both culture regarding touch in general and culture regarding medical use of touch might play a large role, which is not addressed here.

You show that there is no significant difference between object and human touch regarding physical health outcomes. However, I think it need to be addressed that the variance is a lot larger in the non-human touch studies, i.e. that human touch is not simply replacable by non-human touch, but that here probably the exact conditions need to be considered a lot more. This should be addressed in the discussion.

How much later were benefits of touch in newborns assessed? I.e. did you include both immediate effects on effects regarding development evaluated years later? Regarding the latter, if included, the discussion should include the potential limitation that is is not clear whether such effects actually arise from the touch to the child alone or through mediating effects on the relationship/bond between parent (toucher) and child.

Author Rebuttal to Initial comments

Reviewer #1

R1, Point 1:

Overview: The manuscript presents a multivariate meta-analysis and comparative systematic review on the physical and mental health benefits of touch interventions. The work brings together and analyses 137 studies (166 cohorts, 9617 participants, and 643 effect sizes) examining physical and mental health benefits of various types of touch therapies (e.g., massage, kangaroo care, hugging, handholding) for both clinical groups and healthy subjects. By combining various lines of interventions, which target different subject groups (newborn, children, adult, clinical, and healthy cohorts) and by examining the influence of various moderators, the manuscript makes a significant contribution to the scientific understanding of the general physical and mental health benefits of touch interventions. Moreover, there are many highly intriguing results, which could be used in the development of more effective touch interventions and tailoring of touch interventions for specific target groups and clinical/social contexts. For example, the finding that newborns benefit most from a touch provided by a familiar person whereas adults and children benefit similarly from a touch provided by an unfamiliar and familiar sender is highly intriguing and calls for further examination. Similarly, the finding that a robot/object touch has comparable benefits for adult physical health, but poorer mental health outcomes as compared to human-human touch is likewise a result with high clinical/practical relevance.

Response: We thank the reviewer for their appreciation of our work as well as of its significance both for the scientific community and clinical practice. We answer each of the reviewer's comments below and highlight any changes to the manuscript and supplement using the track-changes function.

R1, Point 2:

Validity and robustness of the interpretations: Reading the results, examining the figures, and inspecting all the supplementary files did not bring up flaws that would question the validity or robustness of the main results. The only reservations I had related to the subgroup analyses (and post hoc comparisons) resulting in a large number of statistical tests. Many of the significant p-values tended to stay close to the .05 threshold (e.g. familiarity of toucher in newborns: $p = .041$;

object vs. human touch on adult mental health: $p = .022$). Running many subgroup analyses and pairwise comparisons while assuming a 5% false positive rate is likely to result in some false positives. Can the authors comment on whether and how they handled the type I error risk in the subgroup analyses?

Response: The reviewer raises an important point that is indeed critical to consider when interpreting the results. As the reviewer notes, we investigate a large number of moderators and the risk for type I errors is therefore augmented. One approach to containing false positives was to pre-register all relevant moderators to guide the data analysis process and transparently describe any deviations from the pre-registered plan that potentially increase the risk of alpha errors (Lakens, 2019). While the pre-registration covers the moderators at the primary and secondary level, we however did not specify the procedure regarding the post hoc tests. To reduce risk of inflation here, we initially decided to only run meaningful comparisons, by only investigating differences in, for example, mental health benefits in clinical cohorts to physical health benefits in clinical cohorts. In contrast, a comparison of mental health in clinical cohorts to physical health benefits in healthy cohorts did not seem informative to us. While these measures are helpful, we should nevertheless remain careful when interpreting especially these post hoc findings. Another step to reduce false positive rates was to include random slopes in our random effects structure, a point we elaborate on further under the reviewer's Point 6 as it relates to the suggestion of using the bias-reduced linearization adjustment. Finally, we assumed a rather high correlation between effects (ρ), which the added sensitivity analyses demonstrated to be conservative and thus reduces the likelihood of type I errors (Scammacca et al., 2014). We now revised the discussion section to point at the post hoc tests that are potentially threatened by inflated type I errors. As the reviewer points out, our analyses reveal a number of intriguing findings. Since these results were robust against all sensitivity checks we computed during the revision, we feel that it would serve the field to report these observations. Applying corrections for multiple comparisons systematically would of course have reduced Type I errors effectively, but at the cost of Type II errors, thereby crippling the statistical power in the available literature to the point where reasonable effect sizes would have become undetectable. This would have limited the identification of promising avenues for experimental studies. The new section reads:

Line 387: *"While the majority of findings showed robust health benefits of touch interventions across moderators when compared to a null effect, post hoc tests of, for example, familiarity effects in newborns, or mental health benefit differences between human and object touch, only barely reached significance. Since we computed a large number of statistical tests in the present study, there is a risk that these results are false positives. We hope that researchers in this field are stimulated by these intriguing results and target these questions by primary research through controlled experimental designs within a study."*

R1, Point 3:

Significance: Previous meta-analyses have already demonstrated the health benefits of massage therapy in adults (Moyer et al., 2004; Lee et al., 2015) and kangaroo care in newborns (Boundy et al., 2016). The significance of the current work is that it brings together and enables comparison between these different lines of research while also presenting a larger set of studies with a more extensive set of cohort types, outcomes, and moderators. Therefore, the new meta-analysis provides a more conclusive picture of the health benefits of touch interventions than the previous meta-analyses. Moreover, many of the results may be helpful in developing more effective touch

interventions and when designing haptics and mediated touch technologies.

Response: We thank the reviewer for highlighting the significance of the work for scientists, clinicians, and industry.

R1, Point 4:

Analytical approach: The sample size is also feasible for multivariate meta-analysis. The total sample of studies was further divided into two meta-analyses run separately for the child/adult and the newborn studies, which was justifiable due to the systematic difference between the outcome measures in these two subject groups. To demonstrate sufficient power for subgroup analyses, the authors ran power calculations and examined only those secondary moderators, for which a sufficient number of observations could be found. This, I think, was the right decision due to the large number of sub-group analyses and the limited amount of data for some of the sub-groups.

Response: We thank the reviewer for highlighting the power of our sample size and the justification for separating the sample into two separate groups. We also agree that the approach to calculate a minimum number of effects for subgroup analyses is useful to arrive at more robust conclusions and reduce type I and type II error risks.

R1, Point 5:

The analyses took into account multilevel data structure, the influence of single studies, heterogeneity of the effects sizes, and small study bias. Moreover, the authors analysed the risks of bias in randomization, sequence, performance, and attrition. However, the results of these analyses were not elaborated in the discussion. The same is the case with funnel plots, which showed signs of publication bias as there was a scarcity of smaller studies at the bottom of the funnel. Moreover, the formal test for publication bias was significant for the children/adults set and trending for the newborn set. If understood correctly, the small study bias was taken into account in the analysis by setting the variance of the effect sizes as a moderator in the overarching meta-analyses.

Response: We highly appreciate the reviewer's assessment of the meta-analytic approach to these complex datasets. Our assessment for small study bias does however not directly adjust for bias as such correction models are not yet implemented in multilevel and multivariate meta-analytic approaches. As we mention in the manuscript, we initially aimed to include the RoBMA package that can potentially account for publication bias but this function is not yet suited for complex and hierarchical models which we hope to see implemented in the future. Thus, our reported estimates are likely influenced to some extent by small study bias. We however want to stress that the multivariate reporting of many primary studies in our meta-analyses reduces the risk of publication bias and effect size overestimation considerably. While studies with non-significant findings are often not published, we could extract a large number of non-significant effects from the primary studies since papers often were published if only one out of several outcomes was significant, with plenty of additional null results being reported alongside the main finding. Thus, we believe that publication bias is overall less severe than in meta-analyses without multivariate reporting in primary studies. Furthermore, after carrying out the sensitivity analyses suggested by Reviewer 2, we found that our effect estimates are rather on the conservative side in general. We now elaborated on this finding in detail in the discussion to highlight the presence of

small study bias. Based on a revised analysis of Reviewer 2, there was also small but significant small study bias in the newborn meta-analysis illustrating that the effect estimates might be slightly overestimated. These points are now included in more detail in the manuscript. The new section reads:

Line 393: "Furthermore, the presence of small study bias in both meta-analyses is indicative that the effect size estimates presented here might be overestimated as null results are often unpublished. We want to stress however that this bias is likely reduced by the multivariate reporting of primary studies. Most studies that reported on multiple health outcomes only showed significant findings for one or two amongst many. Thus, the multivariate nature of primary research in this field allowed us to include many non-significant findings in the present study."

R1, Point 6:

From the analysis script, one can see that `robust()` -function of the `metafor` -package was used to provide cluster-robust tests and confidence intervals for the meta-analysis model coefficients. The script indicates that as an adjustment method, the cluster-robust estimate of the variance-covariance matrix was multiplied by the factor $n/(n-p)$ where n refers to the number of clusters and p denotes the total number of model coefficients. Does this method serve as the small-sample adjustment method? Although this adjustment method was applied as a default method by the `robust()` function, it would be good to indicate it in the method section. In the `robust()` function documentation (<https://wviechtb.github.io/metafor/reference/robust.html>), a better small-sample adjustment method is presented utilising a bias-reduced linearization adjustment proposed by Bell and McCaffrey (2002). Can the authors explain why the default method was used instead of this?

Response: We thank the reviewer for highlighting this issue and for asking for clarification on why we used the default method instead of the bias-reduced linearization adjustment method implemented in the `clubSandwich` package. To answer the first question: Yes, the default option in the `robust` function does serve as a small-sample adjustment method (noted in the function documentation (<https://www.rdocumentation.org/packages/metafor/versions/2.4-0/topics/robust>)). Although the bias-reduced linearization might provide a slightly better optimization when the number of effects are small, the `clubSandwich` package does not make it possible to be applied to our model due to our random effects model structure as we included random slopes of the moderator variables at our clustering variable. Random slopes allow our moderators to vary with the random effects which is recommended in the literature on multilevel modelling to reduce the risk of false positives (Oberauer, 2022). We believe that this inclusion is generally beneficial to reduce false positive risk across all models. To ensure the reviewer that this adjustment method however has no effect on any conclusion drawn within the manuscript and only minor effects on numerical values of our meta-analyses, we used a simplified random effects structure without random slopes and compared the default method with the `clubSandwich` adjustment. For the large majority of our data, we saw virtually no difference in the estimation of the confidence intervals. For example, the confidence interval of the overall meta-analysis for adults and children changed the CI from [0.4293; 0.6527] to [0.4292; 0.6529] and the estimate for physical and mental health parameters changed from [0.4417; 0.6912] to [0.4418; 0.6910] and [0.3945;0.6411] to [0.3947;0.6409], respectively. As we rounded to the second digit, applying this adjustment method has no numerical impact on our results. We only saw numerical differences when the number of effects were small ($k < 20$) which was only the case for few subgroup analyses. The post hoc test findings mentioned by the reviewer that we

also discuss in more detail remained significant as well. Therefore, no conclusion drawn in the manuscript was affected by the implementation of this method. We included a table for the reviewer in our revision files comparing results from both adjustment methods. We however believe this table to be beyond the scope of the manuscript due to the different random effects structure resulting in slightly different effect size estimates. Since false positive rates are potentially decreased by the inclusion of random slopes also for estimates with higher number of effects, we would prefer to keep the method as is, since a small-sample adjustment is already implemented in the default function. We are very grateful that the reviewer pointed us at this approach as it provided another robustness check to our data and hope that the reviewer agrees with us here. Since we did not specify the inclusion of random slopes in the original manuscript, we added this information to the methods section together with our reasoning to do so. The new section reads:

Line 586: *"We always included random slopes to allow for our moderators to vary with the random effects at our clustering variable which is recommended in multilevel models to reduce false positives (Oberauer, 2022)."*

R1, Point 7:

Suggested improvements: A more general thing I wanted to comment on is related to the comparative part. That is, the word comparative is in the title but this aspect is pretty much absent in the introduction, methods, and results. In the Results section, there is a small paragraph briefly describing findings regarding the health outcomes of touch in other species. In the discussion, a whole paragraph is used to discuss these findings and their relevance. Thus, there is a clear imbalance between the sections in regard to the comparative aspect. Already in the pre-registration, the lack of animal studies was acknowledged and it was clear that these studies would not be able to be incorporated into the meta-analysis. I consider the health benefits of touching on non-human animals a separate question and out of the scope of the current review/meta-analysis but how do the authors and especially the editor see this?

Response: We agree with the reviewer that due to the sparseness of studies on the benefit of touch for animals, our manuscript mostly focuses on the health benefits of touch interventions for humans. However, in view of the importance of comparative research, and in agreement with the editors, we have kept the small part addressing the animal work within our manuscript. The title now reads: "A systematic review and multivariate meta-analysis of the physical and mental health benefits of touch interventions". We additionally modified the the introduction and methods part to to better reflect the core message of the manuscript. The new sections read:

Line 98: *"Despite the focus of the analysis being on humans, it is widely known that many animal species benefit from touch interactions and that engaging in touch promotes their well-being as well (LaFollette et al., 2018)). Since animal models are essential for the investigation of the mechanisms underlying biological processes and for the development of therapeutic approaches, we accordingly included health benefits of touch interventions in non-human animals as part of our systematic review. However, this search yielded only a small number of studies suggesting a lack of research in this domain and as such, was insufficient to be included in the meta-analysis. We evaluate the identified animal studies and their findings in the discussion."*

Line 459: *"To be included in the systematic review, studies had to investigate the relationship*

between at least one health outcome (physical and/or mental) in humans or animals and a touch intervention (...)

Line 543: *“Due to the lack of identified studies, health benefits to animals were not included as part of the statistical analysis.”*

R1, Point 8:

In my view, in the discussion, one could have elaborated more on the relation of the current meta-analysis results to the results of the previous more limited meta-analyses.

Response: We fully agree with the reviewer and now opened our discussion by highlighting the differences to previous meta-analytic approaches and their limitations in contrast to our present study. The new section reads:

Line 301: *“One limitation of previous meta-analyses is that they focused on specific health outcomes or populations despite primary studies often reporting effects on multiple health parameters simultaneously (e.g., Kong et al. (2013) focusing on neck and shoulder pain; Wang et al. (2013) focusing on massage therapy in preterms). To our knowledge, only Moyer et al. (2004) provided a multivariate picture for a large number of dependent variables. However, this study analyzed their data in separate random effects models that did not account for multivariate reporting nor for the multilevel structure of the data as such approaches have only become recently available. Thus, in addition to adding a substantial amount of new data, our statistical approach provides a more accurate depiction of effect size estimates. Additionally, our study investigated a variety of moderating effects that did not reach significance (e.g., sex ratio, mean age or intervention duration) or were not considered (e.g., the benefits of robot or object touch) in previous meta-analyses in relation to touch intervention efficacy (Moyer et al., 2004), likely because of the small number of studies with information on these moderators in the past. Due to our large-scale approach, we provided high statistical power for many moderator analyses. Finally, previous meta-analyses on this topic exclusively focused on massage therapy in adults or kangaroo care in newborns (Field, 2016) leaving out a large number of interventions that are being carried out in research as well as in everyday life to improve well-being. Incorporating these studies into our study, we found that, in general, both massages and other types of touch, such as gentle touch, stroking or kangaroo care, showed similar health benefits.”*

R1, Point 9:

Moreover, one could have elaborated the results in relation to previous research on how sex and gender influence sensory pleasantness and social comfort associated with touch (Schirmer et al., 2022, Physiology & Behavior) or how attributing tactile stimulation to a social agent (another human vs. robot vs. virtual agent) could boost the mental health benefits of non-skin-to-skin touch (Huisman et al., 2017, IEEE Trans. Hapt Ravaja et al., 2017, Scientific Reports).

Response: This is a very important point as contextual cues are critical for the interpretation and efficacy of touch interventions. We implemented this aspect into our discussion and restructured the paragraph to accommodate these changes by mentioning age-related effects before rather than after the new section. The new section reads:

Line 358: *“For sex differences, our study provides some evidence that there are differences between*

women and men with respect to health benefits of touch. Overall, research on sex differences in touch processing is relatively sparse (but see Russo et al., 2020 and Schirmer et al., 2022). Our results suggest that buffering effects against physiological stress are stronger in women. This is in line with increased buffering effects of hugs in women compared to men (Berretz et al., 2022). The female biased primary research in adults, however, beg for more research in men or non-binary individuals. Unfortunately, our study could not dive deeper in this topic as health benefits broken down by sex or gender were almost never provided. Recent research has demonstrated that sensory pleasantness is affected by sex and that this also interacts with the familiarity of the other person in the touching dyad (Gazzola et al., 2012; Schirmer et al., 2022). In general, contextual factors such as sex and gender or the relationship of the touching dyad, differences in cultural background or internal states, such as stress, have been demonstrated to be highly influential in the perception of affective touch and are thus vital when it comes to maximizing the pleasantness and ultimately the health benefits of touch interactions (Sorokowska et al., 2021; Rajava et al., 2017; Saarinen et al., 2021). As a positive personal relationship within the touching dyad is paramount to induce positive health effects, future research applying robot touch to promote well-being should therefore not only explore synthetic skin options but also improving robots as social agents that form a close relationship with the person receiving the touch (Huisman, 2017)."

R1, Point 10:

Finally, one could have pointed out the risk of alpha inflation in the subgroup analyses as a limitation in the discussion.

Response: We fully agree with the reviewer that this needs to be discussed. As outlined under Point 2, we now added this point to the discussion to clearly delineate that observations that only barely achieved significance need to be revisited in future primary studies. We hope that our study guides researchers in this direction as well-powered experiments will be able to illuminate on these critical findings.

R1, Point 11:

Clarity and context: The manuscript was clearly written and gave sufficient context to the reader to ensure accessibility of the text to the wide readership of Nature Human Behavior. In the abstract, I would have preferred a less technical style of stating the knowledge gap. That is, instead of referring to the effect size and moderators in the second sentence, one could have spoken about these aspects in more layperson terms (e.g., "what factors make touch interventions beneficial for physical and mental health").

Response: We thank the reviewer and agree that the abstract should transmit the knowledge gap and importance of the work in more layperson terms. We have now revised the abstract accordingly. The new section reads:

Lin 39: *"Receiving touch is of critical importance for human well-being. A number of studies have shown that touch promotes mental and physical health. However, what factors influence the health benefits from touch interventions is still awaiting clarification. For example, whether the type of touch, characteristics of the toucher or frequency of touch sessions influences the outcome remains unclear."*

R1, Point 12:

References: I advise going through the list of references carefully because at least Lee et al., 2015 was missing from the list of references.

Response: We apologise for any errors on our part. We now checked all references in the text and the reference list again for completeness.

R1, Point 13:

Conclusion: To conclude, the manuscript presents a large-scale systematic review and meta-analysis of the health benefits of touch interventions with significant contributions to the field. There are a few notions or concerns about some of the secondary moderators being false positives due to alpha inflation. Keeping this risk in mind, the overall results seem robust and significant.

Response: We thank the reviewer again for their appreciation of our work. We believe their comments have improved the readability and clarity of our work, making it more useful for future readers.

References:

- Gazzola, V., Spezio, M. L., Etzel, J. A., Castelli, F., Adolphs, R., & Keysers, C. (2012). Primary somatosensory cortex discriminates affective significance in social touch. *Proceedings of the National Academy of Sciences*, *109*(25), E1657-E1666.
- Huisman, G. (2017). Social touch technology: A survey of haptic technology for social touch. *IEEE transactions on haptics*, *10*(3), 391-408.
- Kong, L. J., Zhan, H. S., Cheng, Y. W., Yuan, W. A., Chen, B. O., & Fang, M. (2013). Massage therapy for neck and shoulder pain: a systematic review and meta-analysis. *Evidence-Based Complementary and Alternative Medicine*, 2013.
- LaFollette, M. R., O'Haire, M. E., Cloutier, S., & Gaskill, B. N. (2018). A happier rat pack: the impacts of tickling pet store rats on human-animal interactions and rat welfare. *Applied Animal Behaviour Science*, *203*, 92-102.
- Lakens, D. (2019). The value of preregistration for psychological science: A conceptual analysis. *Japanese Psychological Review*, *62*(3), 221-230.
- Oberauer, K. (2022). The importance of random slopes in mixed models for Bayesian hypothesis testing. *Psychological Science*, *33*(4), 648-665.
- Ravaja, N., Harjunen, V., Ahmed, I., Jacucci, G., & Spapé, M. M. (2017). Feeling touched: Emotional modulation of somatosensory potentials to interpersonal touch. *Scientific reports*, *7*(1), 40504.
- Saarinen, A., Harjunen, V., Jasinskaja-Lahti, I., Jääskeläinen, I. P., & Ravaja, N. (2021). Social touch experience in different contexts: A review. *Neuroscience & Biobehavioral Reviews*, *131*, 360-372. <https://doi.org/10.1016/j.neubiorev.2021.09.027>
- Scammacca, N., Roberts, G., & Stuebing, K. K. (2014). Meta-analysis with complex research designs: Dealing with dependence from multiple measures and multiple group comparisons. *Review of Educational Research*, *84*(3), 328-364.
- Schirmer, A., Cham, C., Zhao, Z., Lai, O., Lo, C., & Croy, I. (2022). Understanding sex differences in affective touch: Sensory pleasantness, social comfort, and precursive experiences. *Physiology & Behavior*, *250*, 113797.
- Sorokowska, A., Saluja, S., Sorokowski, P., Frąckowiak, T., Karwowski, M., Aavik, T., ... & Croy, I.

(2021). Affective interpersonal touch in close relationships: A cross-cultural perspective. *Personality and Social Psychology Bulletin*, 47(12), 1705-1721.

Wang, L., He, J. L., & Zhang, X. H. (2013). The efficacy of massage on preterm infants: a meta-analysis. *American journal of perinatology*, 731-738.

Reviewer #2

R2, Point 1:

This study conducted a systematic review and a large-scale meta-analysis involving 137 studies with healthy participants and patients to address the benefits of touch on mental and physical health. My research area is meta-analysis, so I will mainly comment on the methodology.

Response: We thank the reviewer for their comments. We answer each of the reviewer's comments below and highlight any changes to the manuscript and supplement using the track-changes function.

R2, Point 2:

1. Lines 196-198. Hedges' g is the primary effect size. However, it is mentioned that effect sizes based on frequency tables are also included in the meta-analysis. How many studies are in frequency tables? What formulas and assumptions for transforming odds ratios to Hedges' g ?

Response: We thank the reviewer for pointing this out. Effect sizes that were calculated from frequency tables were exceedingly rare (11 out of 643 effects). To compute effect sizes from the extracted data, we used the following website (<https://www.campbellcollaboration.org/escalc/html/EffectSizeCalculator-SMD-main.php>).

This website offers a dynamic and easily accessible way to compute effect sizes coming from highly diverse test statistics including p -values, t -values, F -values, means and standard deviations, but also frequency tables. The advantage of this website is that all effect sizes are automatically converted to standardised mean differences (in this case Cohen's d). Thus, odds ratios were never directly calculated in the first place. We used Hedges' formula (1981) to transform Cohen's d to Hedges' g :

$$\text{Hedges' } g = d * \left(1 - \frac{3}{4 * (n_{exp} + n_{control} - 2) - 1}\right)$$

We now added this information to the manuscript:

Line 530: *"The website, <https://www.campbellcollaboration.org/escalc/html/EffectSizeCalculator-SMD-main.php>, was used to convert descriptive and statistical values available in the included studies (means and standard deviations/standard errors/confidence intervals, sample sizes, F -values, t -values, t -test p -values, or frequencies) into Cohen's d , which were then converted in Hedges' g ".*

R2, Point 3:

2. Lines 213-214. $\rho=0.6$ is assumed in calculating the variance-covariance matrix. Since this value may impact the results, it is necessary to conduct a sensitivity analysis by using different values and see if the results are robust.

Response: The reviewer raises an important point. We now conducted sensitivity analyses for values of 0, 0.2, 0.4 and 0.8 in addition to 0.6, for every reported measure in the manuscript. The results of this sensitivity analysis can be found in the Table “Sensitivity analyses” in OSF (<https://osf.io/fs9w2>). As described, the sensitivity analyses demonstrated that effect size estimates across models decrease with higher value of ρ indicating that our initially chosen value can be regarded as conservative (Scammacca et al., 2014). These effect size differences between $\rho = 0$ and $\rho = 0.8$ are however numerically small (see Supplementary Figure 19 and 20 for adults and newborns, respectively). Importantly, no conclusion reported in the manuscript changes by altering ρ , supporting the robustness of our findings. In fact, changing ρ only had an impact on a single variable as effects for respiratory symptoms in adults and children are non-significant at values of $\rho = 0.6$ (as reported in the manuscript) and $\rho = 0.8$ but become significant at lower values of ρ . Apart from this, no significance levels are affected. We thank the reviewer for this important suggestion. We added information on the sensitivity analyses in the methods section, which now reads:

Line 555: *“As ρ needed to be assumed, sensitivity analyses for all computed effect estimates were conducted using correlations between effects of 0, 0.2, 0.4 and 0.8. The results of these sensitivity analyses can be found on the OSF in the Supplementary Table “Sensitivity analyses” (<https://osf.io/fs9w2>). No conclusion drawn in the present manuscript was altered by changing the level of ρ . The sensitivity analyses however showed that higher assumed correlations lead to more conservative effect size estimates (see Supplementary Figures 19 and 20 for the adult and newborn meta-analyses, respectively) reducing type I error risk in general (Scammacca et al., 2014).”*

Figure S19. Influence of different levels of ρ across all computed effect estimates for categorical moderators in the adults meta-analysis. P-values of post hoc tests are depicted whenever significant. Assuming lower levels of correlation between effect sizes resulted in slightly higher effect estimates (repeated measures ANOVA). Thus, higher values of ρ can be considered as conservative estimates. No significant difference was observed between high correlation values of ρ at 0.6 (used value in the manuscript) and 0.8. Effect estimates and confidence intervals for each computed outcome variable can be found on OSF in the Supplementary Table “Sensitivity analyses”.

Figure S20. Influence of different levels of rho across all computed effect estimates for categorical moderators in the newborn meta-analysis. P-values of post hoc tests are depicted whenever significant. Identically to the results for the adults meta-analysis, assuming lower levels of correlation between effect sizes resulted in slightly higher effect estimates (repeated measures ANOVA). No significant difference was observed between high correlation values of rho at 0.6 (used value in the manuscript) and 0.8. Effect estimates and confidence intervals for each computed outcome variable can be found on OSF in the Supplementary Table "Sensitivity analyses".

R2, Point 4:

3. Lines 221-222. The sampling variance of the effect size is used as a moderator in assessing small study bias. Using standard error (SE) is more popular. Can you elaborate on why the sampling variance, not SE, is used?

Response: We thank the reviewer for pointing this out. Since tools like Egger's test are not implemented yet for rma.mv, any measure of effect size precision would theoretically be valid in a meta-regression model, including the sampling variance, the standard error or the inverse sampling variance. We fully agree with the reviewer that it is more standard in the field to use the standard error, and standard errors have been used in simulation studies as well to approximate Egger's regression in multilevel models (Rogers & Pustejovsky, 2020). Since the funnel plots also depict standard errors, a comparison between visual asymmetry identification and the statistical assessment of small study bias is more meaningful if the approaches are parallelized. We therefore decided to change the analysis approach and used the standard error instead. As a result, a test for small study bias became nominally significant where it was only a statistical trend before. We changed the result section accordingly and also added sections in the discussion on the presence of small study bias in our data as requested by the other reviewers. It reads:

Line 393: *"Furthermore, the presence of small study bias in both meta-analyses is indicative that the effect size estimates presented here might be overestimated as null results are often unpublished. We want to stress however that this bias is likely reduced by the multivariate reporting of primary studies. Most studies that reported on multiple health outcomes only showed significant findings for one or two amongst many. Thus, the multivariate nature of primary research in this field allowed us to include many non-significant findings in the present study."*

R2, Point 5:

4. Lines 223-231. Post-hoc power is calculated. It is generally accepted that post-hoc power analysis is not useful. If the observed data is statistically significant, the post-hoc power is large. On the other hand, the post-hoc power is low when the observed data is not statistically significant. I suggest dropping the post-hoc power calculation. If the authors want to keep it, please provide the justifications.

Response: We thank the reviewer for allowing us to further clarify our methodology choice regarding the power calculations. For the subgroup analysis, we believe that looking at subgroups with too few effects for analysis is not particularly informative as non-significant results may simply be underpowered, whereas significant results might be spurious. We however agree that justification is necessary why we believe that the power calculation for the subgroup analysis is still a good approach despite using the effect size estimate from the meta-analysis in a post hoc manner. One critical aspect to consider in terms of publication bias is that many primary studies of our meta-analyses provided multiple reported outcomes, often only few being significant. Thus, a large number of non-significant effects are included in the present effect estimates due to

this common practice in the field. Thus, we believe our post hoc power estimate not to be a large overestimation of the true effect. As an alternative, we also tried to use a medium effect size ($d = 0.5$) as a minimum effect size of interest. We believe that choosing $d = 0.5$ is a reasonable adjustment for the publication bias in the field as reported effects in Psychology are usually in the medium to large range (Szucs et al., 2017) and is likely also a minimal effect size of interest for clinical practice. To achieve at least 80% power, the minimum number of effects only change marginally from $k = 9$ to 10 or $k = 8$ to 9 in the adults and newborn meta-analysis, respectively. This only leads to the exclusion of cortisol data from the outcome subgroup analysis and thus has barely any impact on the manuscript as is. If the reviewer feels that this adjustment is necessary, we are happy to comply with the reviewer's suggestion. The new section reads:

Line 572: "Prior to any sub-group analysis, the overall effect size was used as input for power calculations. While such post hoc power calculations might be limited, we believe that a minimum number of effects to be included in subgroup analyses was necessary to allow for meaningful conclusions. Such medium effect sizes would also likely be the minimum effect sizes of interest for researchers as well as clinical practitioners."

Additionally, the purpose of the analysis presented in the firepower plots was to determine the range of effect sizes that can be reliably detected assuming a range of hypothetical 'true' effect sizes (Lakens, 2022). As the overall effect size from the meta-analysis might be inflated, as evident by the presence of a small study bias, it should be merely used as a useful reference point for the hypothetical true effect size. By presenting power for a range of effects, readers can make up their own minds for what they believe to be important effect size ranges to calculate sample sizes for their future studies. Moreover, using the metameta package is also relevant for helping to "address Item 15 in the Preferred Reporting Items for Systematic Reviews and Meta-Analyses (PRISMA) 2020 checklist—methods used to assess confidence in the body of evidence (Page et al., 2021)—when reporting meta-analyses." (Quintana, 2023). The new section reads:

Line 114: "Using these overall effect estimates, we conducted a power sensitivity analysis of all included primary studies to investigate if such effects could be reliably detected (Lakens, 2022). Sufficient power to detect such effect sizes was rare in individual studies as investigated by firepower plots (Quintana, 2023, see Supplementary Figure S1 and S2)."

R2, Point 6:

5. There are no heterogeneity estimates in the results and the figures. Is it because heterogeneity estimates are not available in robust variance estimation? If yes, please explain this limitation.

Response: We apologise for this lack of clarity as we computed the heterogeneity Q statistic for models in our meta-analyses but only provided heterogeneity estimates in the top left of the orchard or forest plots. This overall test for heterogeneity is significant across all models ($p < .001$) suggesting that heterogeneity is present in the meta-analyses. This is to be expected given the large variety of interventions applied in this field across highly diverse sample cohorts. For multilevel models, heterogeneity indices are indeed different as they are estimated for all random effect levels separately. This indicator (σ^2) is identical to τ^2 in single level random-effects models (Harrer et al., 2021). We now uploaded the Table "Model estimates" to the OSF project that contains the overall Q statistic as well values of σ^2 for each random effects level and refer to it in the methods section:

Line 567: "Heterogeneity in the present study was assessed using Cochran's Q , which determines whether the extracted effect sizes estimate a common population effect size. We also quantified the heterogeneity estimator σ^2 for each random-effects level. These indicators for all models can be found on the OSF in the Table "Model estimates"."

R2, Point 7:

6. In the spreadsheet "AdultsChildren Final datasheet") of the data file "Data final.xlsx," there is a formula (column T) to correct the small bias of d (column S) to Hedges' g . However, there is no formula in the sampling variance (column U). Thus, it is unclear whether the sampling variance in column U is for d or Hedges' g . Please clarify.

Response: We thank the reviewer for pointing this out. In fact, the sampling variance in column U applies for both Cohen's d and Hedges' g . After extracting values of Cohen's d (rounded to two digits), we transformed our effect sizes to Hedges' g to accommodate for small sample bias. As the variance was computed for Cohen's d , we also assumed that a variance correction would be appropriate. However, the variance of the effect sizes is much more dependent on the underlying sample sizes rather than the actual effect sizes. For example, the study of Bennett et al. (2015, adults meta-analysis) provided four effect sizes with 18 participants in the experimental and 18 participants in the control group. Variances for effects of $d = -0.83$ and $d = -0.61$ are $v = 0.12$ and variances for effects of $d = -0.02$ and $d = -0.35$ are $v = 0.11$. The transformation from Cohen's d to Hedges' g only provides mild adjustments that even in extreme cases (high effect size, very low sample size) exceeded differences of $es = 0.1$ only twice among 643 effects. In over 80% of cases, adjustments were in the range of 0.00 and 0.02. None of these adjustments affect the variance meaningfully. Thus, we did not deem a variance correction necessary as the underlying sampling variances were essentially identical. We confirmed this observation by computing the variances with the `escalc` function (metafor R package) for a subset of our data and found identical values for the variances.

References:

- Harrer, M., Cuijpers, P., Furukawa, T., & Ebert, D. (2021). *Doing meta-analysis with R: A hands-on guide*. Chapman and Hall/CRC.
- Hedges, L. V. (1981). Distribution Theory for Glass's Estimator of Effect size and Related Estimators. *Journal of Educational Statistics*, 6(2), 107-128.
- Lakens, D. (2022). Sample size justification. *Collabra: Psychology*, 8(1), 33267.
- Quintana, D.S. (2023). A Guide for Calculating Study-Level Statistical Power for Meta-Analyses. *Advances in Methods and Practices in Psychological Science*, 6(1). <https://doi.org/10.1177/25152459221147260>
- Rodgers, M. A., & Pustejovsky, J. E. (2021). Evaluating meta-analytic methods to detect selective reporting in the presence of dependent effect sizes. *Psychological Methods*, 26(2), 141-160. <https://doi.org/10.1037/met0000300>
- Scammacca, N., Roberts, G., & Stuebing, K. K. (2014). Meta-analysis with complex research designs: Dealing with dependence from multiple measures and multiple group comparisons. *Review of Educational Research*, 84(3), 328-364.
- Szucs D. & Ioannidis J.P.A. (2017). Empirical assessment of published effect sizes and power in the recent cognitive neuroscience and psychology literature. *PLoS Biol* 15(3), e2000797. <https://doi.org/10.1371/journal.pbio.2000797>

Reviewer #3

R3, Point 1:

Packheiser & Hartmann et al. present a timely and important meta-analysis of potential health benefits of touch interventions. Across a large body of studies they find medium-sized beneficial effects of touch on both mental and physical health outcome measures, which they analyse with regard to sub-populations or factors whenever statistically appropriate.

This is an interesting and important work summarising the somewhat hard to compare literature on touch benefits. I only have minor comments:

Response: We thank the reviewer for their appreciation of our work. We answer each of the reviewer's comment below and highlight any changes to the manuscript and supplement using the track-changes function.

R3, Point 2:

Methods: What was the time range of the literature included in the search?

Response: We thank the reviewer for highlighting this missing piece of information, which we have now included in the manuscript.

Line 475: *"In terms of the time frame, we conducted an open-start search of studies until 2022 and identified studies conducted between 1965 and 2022."*

R3, Point 3:

Language of the included articles is unclear. You write French, Dutch, German, English in the text, but the table says some were excluded for not being in English.

Response: We apologise for the confusion and have updated the flowchart in Figure 1 to reflect the correct inclusion criterion, which was articles in French, Dutch, German, or English. However, we did not identify any articles in French, Dutch or German using our search criteria, which we detail in the caption of Figure 8:

Caption Figure 8: *"Included languages were French, Dutch, German, and English, however, our search did not identify any articles in French, Dutch, or German."*

R3, Point 4:

How were the specific outcome measures chosen?

Response: We thank the reviewer for allowing us to clarify this point. We decided to be inclusive with respect to outcome measures and chose all measures that were related to mental and physical health aspects. For specific health outcomes, we also only investigated them further if a sufficient number of effects were available as per our power analysis. Measures such as

neurophysiological responses from for example EEG recordings or pleasantness ratings to touch

were excluded as they do not reflect any health outcomes. We explain some of these decisions in our deviations from the preregistration:

Line 428: *“Deviation #1: During our initial screening for the systematic review, we were confronted with a large number of potential health outcomes to look at. This observation of multivariate outcomes led us to register an amendment during data collection (but prior to any effect size or moderator screening). In doing so, we aimed to additionally extract meta-analytic effects for a more quantitative assessment of our review question that can account for multivariate data reporting and dependencies of effects within the same study.”*

We have also added the following additional information in the manuscript:

Line 465: *“Regarding the included health outcomes, we aimed to be as broad as possible but excluded parameters such as neurophysiological responses or pleasantness ratings after touch application as they do not reflect health outcomes. We then applied a two-step process. First, we identified all potential health outcomes and extracted qualitative information on those outcomes (e.g., direction of effect). Second, we extracted quantitative information from all possible outcomes (e.g. effect sizes).”*

R3, Point 5:

For the meta-analyses, only between-subject designs were included, however, in the bias analysis you mention both between- and within-designs. Please clarify.

Response: We thank the reviewer for allowing us to clarify this point. For the systematic review, we aimed to broadly include both between- and within-subject designs so as to assess the full picture of prior work. As risk of bias analyses are an integral part of systematic reviews and meta-analyses, we conducted this analysis on all study designs and included studies of both the review and the meta-analyses. However, for the meta-analysis, we excluded within-subject designs for two reasons: first, from a conceptual point of view, we wanted to clearly distinguish touch from no-touch effects since a between-subject design ensures that there was no touch intervention in the control group. The second reason was of methodological nature as we had no information about the correlation between measurements that are important for estimating effect estimates in within-subject designs (Lakens, 2013). We have now clarified this reasoning in the manuscript:

Line 595: *“For a broad overview of prior work and their biases, risk of bias was assessed for all studies included in both meta-analyses and the systematic review.”*

Line 470: *“The meta-analysis additionally required a between-subjects design (in order to clearly distinguish touch from no-touch effects and due to missing information about the correlation between repeated measurements (Lakens, 2013)).”*

It is of note that within-subject designs were rare (11 studies in total) and an exclusion from the systematic review would not alter any conclusions from this part of our study.

R3, Point 6:

Also regarding the bias, you include not being blinded in the parameters, however, I assume it not to be possible to be blinded to conditions in a touch vs. no touch study.

Response: We completely agree with the reviewer, and have highlighted this information in the manuscript (which is also again mentioned in the Risk of Bias Figure S21):

Line 612: *"Importantly, performance bias was always high in the adult/children meta-analysis as blinding of the participants and experimenters to the experimental conditions was not possible due to the nature of the intervention. For studies with newborns and animals, we assessed the performance bias as medium since neither newborns or animals are likely to be aware of being part of an experiment or specific group."*

R3, Point 7:

Where did you draw the line between human and object/robot touch? Only skin-to-skin vs. all other? What about e.g. handheld tools (brushes) or what about touch that is not skin-to skin (through clothing or with gloves)?

Response: We thank the reviewer for bringing up this important point. We considered all touch as human where a human physically touched the recipient either with or without skin-to-skin contact (e.g., also with clothes or gloves on). In contrast, object or robot touch was defined as any touch where no human was involved (e.g. vibration devices). No study used brushes to improve health as this type of methodology seems exclusive for studying pleasantness or brain responses to touch which we excluded for not reflecting health outcomes. For both the newborn and the adult/children samples, all studies classified as human touch included actual human touch and no aids such as brushes or tools. We now clarified this in the manuscript:

Line 464: *"Human touch was always explicit touch by a human (i.e., no brushes or other tools), either with or without skin-to-skin."*

R3, Point 8:

Results/Discussion: While I appreciate your clean and fact based style, I think the article would benefit from a few examples and descriptors of both the touch interventions and the control conditions, e.g. the most common types used.

Response: We thank the reviewer and agree that the article would benefit from more examples to clarify our investigated concepts. We have now revised the manuscript accordingly:

Line 39: *"A number of studies have shown that touch promotes mental and physical health. However, what factors influence the health benefits from touch interventions is still awaiting clarification. For example, whether the type of touch, characteristics of the toucher or frequency of touch sessions influences the outcome remains unclear."*

Line 74: *“The most common touch interventions, for example massage for adults or kangaroo care for newborns, have been shown to have a wide range of both mental and physical health benefits (...)”*

Line 515: *“Common control groups for touch interventions comprised active (e.g., relaxation therapy) as well as passive control groups (e.g., standard medical care).”*

Line 77: *“Despite the substantial weight this literature gives to support the benefits of touch, it is also characterized by a large variability in, for example, studied cohorts (adults, children, newborns, animals), type and duration of applied touch (e.g., one-time hug vs. 60-min massage), measured health outcomes (ranging from physical health outcomes such as sleep and blood pressure to mental health outcomes such as depression or mood), and who actually applies the touch (e.g., partner vs. stranger).”*

We also added a Supplementary Table to the Supplementary Information. A more detailed list and information on single studies can be found here in the corresponding Open Science Framework project.

Table S1. List of touch interventions of touched body parts across the included studies

List of touch interventions	List of touch body parts
Massage therapy (including self-massage)	Head (including face)
Kangaroo care (including skin-to-skin contact) for newborns only	Arm (including hands) Leg (including feet)
Other touch (including gentle touch, Yakson touch, maternal touch, tactile and kinesthetic stimulation, haptic touch, vibration touch, superficial touch, object touch, physical touch, self touch, stroking, rocking, fondling, petting, patting, acupressure, hugging, close contact, gentle touch, passive movement, child touch, kangaroo (for adults/children only))	Torso (including back, chest, stomach and neck) Multi (whenever multiple parts of the body were touched, e.g. full-body massages and all studies where newborns were touched)

R3, Point 9:

What was the standard deviation of age in the meta analysis sample?

Response: The standard deviation of mean age was 21.16 years. We added this information to the manuscript:

Line 249: *“The median age in the adult meta-analysis was 42.6 years (SD = 21.16; range: 4.5 - 88.4).”*

R3, Point 10:

Could you include a list of all touched body parts and of all types of touch interventions in the supplement?

Response: We agree that this is a valuable addition and have added a Supplementary Table to the Supplementary Information. A more detailed list and information on single studies can be found here in the corresponding Open Science Framework project.

Table S1. List of touch interventions of touched body parts across the included studies

List of touch interventions	List of touch body parts
Massage therapy (including self-massage)	Head (including face)
Kangaroo care (including skin-to-skin contact) for newborns only	Arm (including hands) Leg (including feet)
Other touch (including gentle touch, Yakson touch, maternal touch, tactile and kinesthetic stimulation, haptic touch, vibration touch, superficial touch, object touch, physical touch, self touch, stroking, rocking, fondling, petting, patting, acupressure, hugging, close contact, gentle touch, passive movement, child touch, kangaroo (for adults/children only))	Torso (including back, chest, stomach and neck) Multi (whenever multiple parts of the body were touched, e.g. full-body massages and all studies where newborns were touched)

R3, Point 11:

Animal studies: are these only touches by humans to the animal or also touch by members of the same species?

Response: Animal studies included both human and object touch interventions. While animal-to-animal touch is of high interest and takes place for example in consolation paradigms (Wu et al, 2020), we noticed the difficulties to conduct an intraspecies intervention under controlled conditions in these studies. Unfortunately, no such study complied with the inclusion criteria that were set for this manuscript. The results and methods part have been amended to clarify this point in the manuscript. The new section reads as follows:

Line 290: *“We thus found strong evidence that touch interventions, which were mostly conducted by humans (16 studies human touch, 3 studies object touch), had positive health effects in animal species as well.”*

Line 462: *“Of note, as a result of this selection process, no animal-to-animal touch intervention studies were included as they never featured a proper no-touch control.”*

R3, Point 12:

Discussion: While it is to be assumed that most scientific peer-reviewed studies are published in English language, the specific field of research of touch benefits on health outcomes might also be addressed in a larger body of articles in languages other than English, considering that manual practices and massage therapy are actually more common in countries like China and India. This might be a limitation, as these studies were not included. In addition, both culture regarding touch in general and culture regarding medical use of touch might play a large role, which is not addressed here.

Response: We fully agree with the reviewer. Although a considerable body of evidence from Asian countries such as India, China or Japan was included, many more written in their native languages might exist. We added this as a limitation:

Line 399: *“Another limitation pertains to the fact that we only included articles in languages mostly spoken in Western countries. As a large body of evidence comes from Asian countries, it could be that primary research was published in languages other than specified in the inclusion criteria. Despite the large and inclusive nature of our study, some studies could thus have been missed.”*

We also included cultural differences in the discussion as an important aspect for consideration in this line of research. It reads:

Line 366: *“In general, contextual factors such as sex and gender or the relationship of the touching dyad, differences in cultural background, or internal states such as stress have been demonstrated to influence the perception of affective touch and are thus relevant to maximizing the pleasantness and ultimately the health benefits of touch interactions (Sorokowska et al., 2021; Rajava et al., 2017; Saarinen et al., 2021).”*

R3, Point 13:

You show that there is no significant difference between object and human touch regarding physical health outcomes. However, I think it need to be addressed that the variance is a lot larger in the non-human touch studies, i.e. that human touch is not simply replacable by non-human touch, but that here probably the exact conditions need to be considered a lot more. This should be addressed in the discussion.

Response: The reviewer rightfully points out this important point. We now added this aspect to the discussion. The new sections read:

Line 339: *“It should be noted that, although we did not observe significant differences in physical health benefits between human-human and human-object touch, the variability of effect sizes was higher in human-object touch. The conditions enabling object or robot interactions to improve well-being should therefore be explored in more detail in the future.”*

Line 370: *“As a positive personal relationship within the touching dyad is paramount to induce positive health effects, future research applying robot touch to promote well-being should therefore not only explore synthetic skin options but also improving robots as social agents that form a close relationship with the person receiving the touch (Huisman, 2017).”*

R3, Point 14:

How much later were benefits of touch in newborns assessed? I.e. did you include both immediate effects on effects regarding development evaluated years later? Regarding the latter, if included, the discussion should include the potential limitation that is is not clear whether such effects actually arise from the touch to the child alone or through mediating effects on the relationship/bond between parent (toucher) and child.

Response: We thank the reviewer for this important comment. Indeed, we only extracted immediate effects (health outcome assessed directly after the intervention), our reasoning

being that long-term effects are not necessarily directly (or only) associated with the initial touch intervention. The only exceptions were death rate or hospital stay duration in newborns that however were still relatively close to the intervention and could be clearly attributed to the intervention due to the controlled conditions in the hospital. This information is now reflected in the manuscript:

Line 508: *"If multiple time points were measured for the same outcome on the same day after a single touch intervention, we extracted the peak effect size (in either the positive or negative direction). If the touch intervention occurred multiple times and health outcomes were assessed for each time point, we extracted data points separately. However, we only extracted immediate effects, as long-term effects not controlled through the experimental conditions could be due to influences other than the initial touch intervention."*

Line 413: *"Similarly, it is important to note that our conclusions mainly refer to outcomes measured close to the touch intervention, as we did not include long-term outcomes years after."*

References:

- Lakens, D. (2013). Calculating and reporting effect sizes to facilitate cumulative science: a practical primer for t-tests and ANOVAs. *Frontiers in Psychology*, 4, 863.
- Huisman, G. (2017). Social touch technology: A survey of haptic technology for social touch. *IEEE transactions on haptics*, 10(3), 391-408.
- Ravaja, N., Harjunen, V., Ahmed, I., Jacucci, G., & Spapé, M. M. (2017). Feeling touched: Emotional modulation of somatosensory potentials to interpersonal touch. *Scientific reports*, 7(1), 40504.
- Saarinen, A., Harjunen, V., Jasinskaja-Lahti, I., Jääskeläinen, I. P., & Ravaja, N. (2021). Social touch experience in different contexts: A review. *Neuroscience & Biobehavioral Reviews*, 131, 360–372.
- Sorokowska, A., Saluja, S., Sorokowski, P., Frąckowiak, T., Karwowski, M., Aavik, T., ... & Croy, I. (2021). Affective interpersonal touch in close relationships: A cross-cultural perspective. *Personality and Social Psychology Bulletin*, 47(12), 1705-1721.

Decision Letter, first revision:

20th December 2023

Dear Dr. Packheiser,

Thank you for your patience as we've prepared the guidelines for final submission of your Nature Human Behaviour manuscript, "A systematic review and multivariate meta-analysis of the physical and mental health benefits of touch interventions" (NATHUMBEHAV-23082675A). Please carefully follow the step-by-step instructions provided in the attached file, and add a response in each row of the table to indicate the changes that you have made. Please also address the additional marked-up edits we have proposed within the reporting summary. Ensuring that each point is addressed will help to ensure that your revised manuscript can be swiftly handed over to our production team.

We would hope to receive your revised paper, with all of the requested files and forms within two-three weeks. Please get in contact with us if you anticipate delays.

If you have not done so already, please alert us to any related manuscripts from your group that are under consideration or in press at other journals, or are being written up for submission to other journals (see:

<https://www.nature.com/nature-research/editorial-policies/plagiarism#policy-on-duplicate-publication> for details).

Nature Human Behaviour offers a Transparent Peer Review option for new original research manuscripts submitted after December 1st, 2019. As part of this initiative, we encourage our authors to support increased transparency into the peer review process by agreeing to have the reviewer comments, author rebuttal letters, and editorial decision letters published as a Supplementary item. When you submit your final files please clearly state in your cover letter whether or not you would like to participate in this initiative. Please note that failure to state your preference will result in delays in accepting your manuscript for publication.

In recognition of the time and expertise our reviewers provide to Nature Human Behaviour's editorial process, we would like to formally acknowledge their contribution to the external peer review of your manuscript entitled "A systematic review and multivariate meta-analysis of the physical and mental health benefits of touch interventions". For those reviewers who give their assent, we will be publishing their names alongside the published article.

Cover suggestions

We welcome submissions of artwork for consideration for our cover. For more information, please see our https://www.nature.com/documents/Nature_covers_author_guide.pdf target="new"> guide for cover artwork.

ORCID

Non-corresponding authors do not have to link their ORCIDs but are encouraged to do so. Please note that it will not be possible to add/modify ORCIDs at proof. Thus, please let your co-authors know that if they wish to have their ORCID added to the paper they must follow the procedure described in the following link prior to acceptance:

Nature Human Behaviour has now transitioned to a unified Rights Collection system which will allow our Author Services team to quickly and easily collect the rights and permissions required to publish your work. Approximately 10 days after your paper is formally accepted, you will receive an email in providing you with a link to complete the grant of rights. If your paper is eligible for Open Access, our Author Services team will also be in touch regarding any additional information that may be required to arrange payment for your article.

Please note that *Nature Human Behaviour* is a Transformative Journal (TJ). Authors may publish their research with us through the traditional subscription access route or make their paper immediately open access through payment of an article-processing charge (APC). Authors will not be required to make a final decision about access to their article until it has been accepted. Find out more about Transformative Journals

[REDACTED]

Best regards,
Alex McKay
Editorial Assistant
Nature Human Behaviour

On behalf of

Samantha Antusch

Samantha Antusch, PhD
Senior Editor
Nature Human Behaviour

Reviewer #1:

Remarks to the Author:

Revisions made to the manuscript and the authors' responses to reviewers' comments were

thorough and well-motivated. The authors defended their methodological and technical choices adequately and ran substantial amounts of sensitivity analyses and other checks providing further evidence for the robustness of their results.

Specifically, in response to Reviewer 2, sensitivity analysis was conducted to examine the impact of different rho values on the main results demonstrating the selected rho=0.6 to be conservative. In response to my question about why the default small sample adjustment was used instead of bias-reduced linearization adjustment, the authors pointed out that it was not possible because the moderator variables were specified as random slopes. To demonstrate further how using the bias-reduced linearization adjustment had no notable effect on the results, the authors ran the models again without random slopes and using the clubSandwich adjustment instead. I'm satisfied with this response.

Furthermore, I pointed out the risk that some of the sub-group analysis results could be false positives due to alpha inflation. In response, the authors raised this limitation in the discussion while stressing the importance of reducing type-II error and the chance for new discoveries. Regarding the management of type-I error risk, the authors pointed out that the analyses were preregistered, a sensitivity power analysis was carried out using conservative effect-size parameters, and random slopes of the moderator variables were estimated. I agree that these procedures make the results more robust reducing the risk of type-I error.

Finally, I was satisfied with the way the authors handled the imbalance produced by their comparative work in the original manuscript. Although I did find the comparative examination of health benefits in non-human animals less central to the work, I do appreciate that the review was retained in the work.

I want to thank the authors for their thorough responses and revisions and recommend the article to be accepted for publication.

Kind Regards,
Dr. Ville Harjunen

Reviewer #2:

Remarks to the Author:

I appreciate the authors' hard work in addressing my concerns. They have effectively addressed all of my questions. However, after reviewing the revision, I have two additional comments that I hope the authors can address.

Firstly, Cochran's Q test is used to test the homogeneity of effect sizes. However, it assumes that all effect sizes are independent. This may not be the case in the present setting, as the effect sizes may be structured and nested. I wonder if this assumption still holds true. If not, a brief note to remind the readers will be helpful.

Secondly, the meta-analysis model used is multistep, multivariate, and multilevel, with random slopes. The authors utilized the mpower R package to conduct the power analysis. However, my limited knowledge of the mpower R package suggests that it is used for power analysis for primary

data. I am curious as to how the meta-analysis structure was taken into consideration when computing the power analysis.

Reviewer #3:

Remarks to the Author:

The authors addressed all my concerns. I have no further comments.

Author Rebuttal, first revision:

Reviewer #1 (Remarks to the Author):

Point 1:

Revisions made to the manuscript and the authors' responses to reviewers' comments were thorough and well-motivated. The authors defended their methodological and technical choices adequately and ran substantial amounts of sensitivity analyses and other checks providing further evidence for the robustness of their results.

Specifically, in response to Reviewer 2, sensitivity analysis was conducted to examine the impact of different rho values on the main results demonstrating the selected rho=0.6 to be conservative. In response to my question about why the default small sample adjustment was used instead of bias-reduced linearization adjustment, the authors pointed out that it was not possible because the moderator variables were specified as random slopes. To demonstrate further how using the bias-reduced linearization adjustment had no notable effect on the results, the authors ran the models again without random slopes and using the clubSandwich adjustment instead. I'm satisfied with this response.

Furthermore, I pointed out the risk that some of the sub-group analysis results could be false positives due to alpha inflation. In response, the authors raised this limitation in the discussion while stressing the importance of reducing type-II error and the chance for new discoveries. Regarding the management of type-I error risk, the authors pointed out that the analyses were preregistered, a sensitivity power analysis was carried out using conservative effect-size parameters, and random slopes of the moderator variables were estimated. I agree that these procedures make the results more robust reducing the risk of type-I error.

Finally, I was satisfied with the way the authors handled the imbalance produced by their comparative work in the original manuscript. Although I did find the comparative examination of

health benefits in non-human animals less central to the work, I do appreciate that the review was retained in the work.

I want to thank the authors for their thorough responses and revisions and recommend the article to be accepted for publication.

Kind Regards,

Dr. Ville Harjunen

Response: We want to again thank the reviewer for the positive assessment of our manuscript. We are happy that the reviewer was satisfied with our revisions and for the recommendation to publish our manuscript.

Reviewer #2 (Remarks to the Author):

Point 1:

I appreciate the authors' hard work in addressing my concerns. They have effectively addressed all of my questions. However, after reviewing the revision, I have two additional comments that I hope the authors can address.

Response:

We thank the reviewer for the positive feedback. The additional concerns are addressed below.

Point 2:

Firstly, Cochran's Q test is used to test the homogeneity of effect sizes. However, it assumes that all effect sizes are independent. This may not be the case in the present setting, as the effect sizes may be structured and nested. I wonder if this assumption still holds true. If not, a brief note to remind the readers will be helpful.

Response:

We thank the reviewer for the opportunity to clarify on this issue. The Q statistic is computed as part of the nested models in the `rma.mv` function. As such, it considers the dependencies of the effect sizes in the model already. We added this information to the methods section. We attached a quote from the `rma.mv` documentation to the revision letter: “A test for (residual) heterogeneity is automatically carried out by the function. Without moderators in the model, this test is the generalized/weighted least squares extension of Cochran's Q-test, which tests whether the variability in the observed effect sizes or outcomes is larger than one would expect based on sampling variability (and the given covariances among the sampling errors) alone.” The new section reads:

“Although the Q statistic in the `rma.mv` function accounts for the hierarchical nature of the data, we also quantified the heterogeneity estimator σ^2 for each random-effects level to provide a comprehensive overview of heterogeneity indicators.”

Secondly, the meta-analysis model used is multistep, multivariate, and multilevel, with random slopes. The authors utilized the `mpower` R package to conduct the power analysis. However, my limited knowledge of the `mpower` R package suggests that it is used for power analysis for primary data. I am curious as to how the meta-analysis structure was taken into consideration when computing the power analysis.

Response:

We thank the reviewer for highlighting this point and apologize for the confusion. In fact, we used the `mpower` function from the `metapower` package in R and not the `mpower` package which indeed is used for power calculation of primary data. Thus, the meta-analytic nature of the data is part of the power calculation. Power calculations for this meta-analysis followed the guidelines published by Quintana (2023). We corrected the error in the supplementary material and now correctly refer to the `metapower` package (Griffin, 2021).

<https://www.rdocumentation.org/packages/metapower/versions/0.2.2/topics/mpower>

Quintana, D. S. (2023). A guide for calculating study-level statistical power for meta-analyses. *Advances in Methods and Practices in Psychological Science*, 6(1), 25152459221147260.

Griffin, J. W. (2021). Calculating statistical power for meta-analysis using `metapower`. *The Quantitative Methods for Psychology*.

Reviewer #3 (Remarks to the Author):

The authors addressed all my concerns. I have no further comments.

Response:

We want to thank the reviewer for their positive assessment of our revisions and are content that all concerns were addressed.

Final Decision Letter:

Dear Dr Packheiser,

We are pleased to inform you that your Article "A systematic review and multivariate meta-analysis of the physical and mental health benefits of touch interventions", has now been accepted for publication in *Nature Human Behaviour*.

Please note that *Nature Human Behaviour* is a Transformative Journal (TJ). Authors may publish their research with us through the traditional subscription access route or make their paper immediately open access through payment of an article-processing charge (APC). Authors will not be required to make a final decision about access to their article until it has been accepted. Find out more about Transformative Journals

With best regards,

Samantha Antusch

Samantha Antusch, PhD
Senior Editor
Nature Human Behaviour